# CHUNK-DISTILLED LANGUAGE MODELING

**Yanhong Li**
University of Chicago & TTI-Chicago
yanhongli@uchicago.edu

**Karen Livescu**
TTI-Chicago
klivescu@ttic.edu

**Jiawei Zhou**
TTI-Chicago & Stony Brook University
jiawei.zhou.1@stonybrook.edu

## ABSTRACT

We introduce Chunk-Distilled Language Modeling (CD-LM), an approach to text generation that addresses two challenges in current large language models (LLMs): the inefficiency of token-level generation, and the difficulty of adapting to new data and knowledge. Our method combines deep network-based LLMs with a straightforward retrieval module, which allows the generation of multi-token text chunks at a single decoding step. Our retrieval framework enables flexible construction of model- or domain-specific datastores, either leveraging the internal knowledge of existing models, or incorporating expert insights from human-annotated corpora. This adaptability allows for enhanced control over the language model's distribution without necessitating additional training. We present the CD-LM formulation along with performance metrics demonstrating its ability to improve language model performance and efficiency across a diverse set of downstream applications.[1]

## 1 INTRODUCTION

Large language models (LLMs) have become a crucial component of intelligent systems, but still suffer from fundamental challenges to their efficiency and performance. LLMs are most commonly based on autoregressive Transformers (Vaswani et al., 2017) and typically generate text sequences one token at a time in a serial fashion, which limits their efficiency. Moreover, once pre-trained, updating the model parameters requires expensive data and computational resources, making it difficult to incorporate dynamic knowledge into the model.

Several techniques have been proposed to improve the efficiency and performance of LLMs, such as speculative decoding (Leviathan et al., 2023; Chen et al., 2023; Miao et al., 2024; Spector & Re, 2023) and retrieval-augmented generation (RAG) (Lewis et al., 2020; Guu et al., 2020; Borgeaud et al., 2022). The former relies on a smaller model to speculate several tokens at a time to reduce inference runtime while retaining the same model distribution, while the latter combines parametric language models with non-parametric memory to improve adaptability to dynamic knowledge but often without efficiency gains.

This work aims to alleviate both challenges via a fine-grained retrieval-augmented language modeling approach that focuses on text chunks, or contiguous spans of tokens that often appear together. The intuition for this approach is that a substantial amount of linguistic or factual knowledge can be expressed in text chunks spanning multiple contiguous tokens, such as named entities, multi-word expressions, and other common phrases. These sub-sentence structures tend to exhibit lower variability compared to larger text units such as sentences, and are often memorized precisely by well-trained LLMs. Figures 1 and 2 demonstrate this effect: Chunks conveying key content are often repeated verbatim across multiple decoding runs with similar contexts, and the LLM probabilities over token sequences show recurring *plateaus of high probability* within such multi-token chunks. By injecting memorized or novel chunks into the generation process, we may be able to

---

[1]Code and data are available at https://github.com/yanhong-lbh/cd-lm.

| USER | What is the answer to life the universe and everything? | USER | The answer to life, the universe, and everything is? | USER | Tell me the meaning of life, the universe and everything. |
|------|------|------|------|------|------|
| LLM | The answer to life, the universe, and everything is 42, according to Douglas Adams' "The Hitchhiker's Guide to the Galaxy." | LLM | The answer to everything is 42, reflecting Douglas Adams' humorous take on the quest for universal truths in "The Hitchhiker's Guide to the Galaxy." | LLM | The meaning of life, the universe, and everything is 42, referencing Douglas Adams' iconic work in "The Hitchhiker's Guide to the Galaxy." |

Figure 1: LLMs may generate sequences with repeated chunks spanning contiguous tokens conveying key information in similar contexts. Examples are generated from Llama-2-7b-chat.

improve the models' ability to adapt to new domains or knowledge. In addition, if entire chunks can be cached and retrieved during inference, we should also be able to speed up text generation.

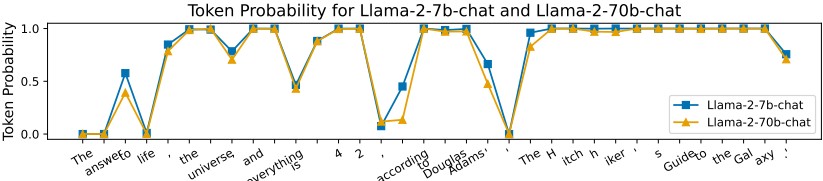

Figure 2: LLM token probabilities for the sentence: "*The answer to life, the universe, and everything is 42, according to Douglas Adams' The Hitchhiker's Guide to the Galaxy.*" These models bind token sequences such as *Douglas Adams'* and *The Hitchhiker's Guide to the Galaxy* into chunks with plateaus of high probability.

Inspired by these observations, we present Chunk-Distilled Language Modeling (CD-LM), a new training-free generation approach that mixes LM token generation with chunk retrieval. To facilitate efficient search, we store text chunks of variable sizes, along with their preceding contexts, in a trie-structured datastore, and retrieve the most likely chunks as possible text continuations given the current generation. The context matching is done in the vector representation space induced by the LM itself without the additional overhead of specialized embedding modules, commonly used in RAG (Lan et al., 2023; Ram et al., 2023; Borgeaud et al., 2022).

Well-matched chunk continuations are accepted, skipping multiple token decoding steps. Using the same generation approach, CD-LM allows language models (LMs) to work with chunks mined in different ways to achieve various goals in applications. As suggested by Figure 2, chunks can be naturally derived from any parametric pre-trained LM as memorized high-probability sequences. When the chunks are extracted from the distribution of a more powerful or specialized LM, CD-LM implements a form of knolwedge distillation, adapting the base model's distribution (without any additional training) by injecting chunks during inference. In this setting CD-LM can either improve smaller models with knowledge drawn from larger models or perform training-free domain adaptation. On the other hand, when the chunks are extracted from the same LM used for generation, they form a self-memory datastore that can be used to improve inference efficiency while maintaining the same model distribution, as in speculative decoding. Finally, the chunks can be not only extracted from a parametric model but even directly curated from human experts. Such external knowledge can be factual information or private data that the LM may not have direct access to.

CD-LM requires no training and can work with any off-the-shelf language model in both chunk discovery and sequence generation. We conduct a diverse set of empirical studies, including language modeling perplexity, text generation, and domain adaptation, showing the ability of CD-LM to improve inference efficiency and modeling performance.

## 2 BACKGROUND

While many attempts have been made to improve language modeling and generation efficiency, it remains a significant challenge to address both simultaneously. For example, non-parametric approaches like kNN-LM (Khandelwal et al., 2020) reduce LM perplexity in certain domains, but tend to require a sizable database for retrieval and adds latency during generation; specialized inference algorithms like speculative decoding (Spector & Re, 2023) speed up generation but keep the LM's distribution fixed. Unlike prior work, CD-LM can both speed up generation and adapt the LM's distribution. We include a more comprehensive overview of related work in Appendix C.

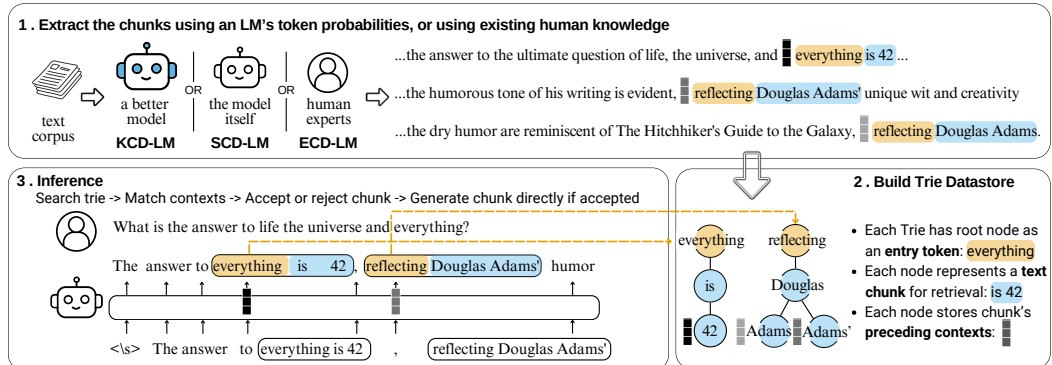

Figure 3: Overview of CD-LM. Colored text spans are generated together by chunk retrieval, interleaved with token-by-token generation by the LM. Note that same chunk can appear in multiple contexts, so each node in the trie datastore contains multiple context vectors in practice.

**Non-Parametric Language Modeling** kNN-LM (Khandelwal et al., 2020) extends a pre-trained LM by linearly interpolating its distribution with a non-parametric k-nearest neighbors model based on token retrieval, thereby often improving language modeling performance. However, it is typically very inefficient as it performs retrieval at each token, and it affects the immediate next token distribution via soft mixing. There is a series of proposed methods that improve the efficiency of kNN-LM (He et al., 2021; Alon et al., 2022); however, they are still slower than the pre-trained LM. Unlike kNN-LM, CD-LM does not involve retrieval at each token and makes a hard decision about multiple tokens in a chunk rather than mixing token distributions, enabling it to enjoy the benefits of dynamic retrieval but also save on kNN searches.

**Speculative Decoding** Speculative decoding (Leviathan et al., 2023; Chen et al., 2023; Miao et al., 2024; Spector & Re, 2023; He et al., 2024) is an inference acceleration technique. Given a particular target LLM, a smaller LM is used to quickly generate multiple draft tokens, which are then considered together by the target LLM. The work most closely related to ours is REST (He et al., 2024), which retrieves draft token sequences from an external datastore. While CD-LM also retrieves chunks from a datastore, it is fundamentally different from speculative decoding. Speculative decoding methods use the target LLM for draft token verification, so the language model's distribution and therefore downstream performance cannot be further improved and no new knowledge can be injected. In contrast, CD-LM is designed to inject chunk-level knowledge into generation so the model distribution can be adapted.

## 3 LANGUAGE MODELING WITH CHUNK GENERATION

In this section, we introduce a general framework of language modeling that interleaves chunk generations with tokens from a standard autoregressive LM. We then describe the operational details of the chunk generation process with retrieval from a structured database in Section 4. Together, these two sections build the core ideas of CD-LM. Finally, we derive a tractable algorithm for computing sequence probabilities under CD-LM in Section 5.

### 3.1 PRELIMINARIES

An autoregressive language model assigns a probability to any given sequence of tokens $(x_1, x_2, \ldots, x_N)$ as follows: $p_\theta(x_1, x_2, \ldots, x_N) = \prod_{n=1}^{N} p_\theta(x_n | x_{<n})$, where $\theta$ is the model parameters and $x_{<n} = (x_1, x_2, \ldots, x_{n-1})$. Modern LLMs are usually parameterized by Transformer (Vaswani et al., 2017) architectures composed of stacks of self-attention and feedforward layers. Individual tokens from a closed vocabulary $V$ are sequentially passed into the model with their embedding vectors, and the next token probability distribution is computed by

$$h_n = f_\theta(x_1, x_2, \ldots, x_{n-1})$$
$$p_\theta(x_n | x_{<n}) = \text{softmax}(W_o h_n) \tag{1}$$

where $f_\theta(\cdot)$ denotes the functional process that maps the previous sequence of tokens into a fixed-size *context vector* $h_n \in \mathbb{R}^d$, and $W_o \in \mathbb{R}^{|V| \times d}$ is the output embedding matrix that projects the

representation vector onto the vocabulary space. Given a learned model, text can be generated by sampling from the next token distribution autoregressively one token at a time, resulting in $N$ forward runs for a sequence of length $N$.

## 3.2 Text Chunk Generation Modeling

Instead of producing text one token at a time, we provide a mechanism that can directly generate a span of multiple consecutive tokens, or chunks, with better efficiency and flexibility of injecting knowledge on fine-grained sub-sentence levels into the model distribution on the fly.

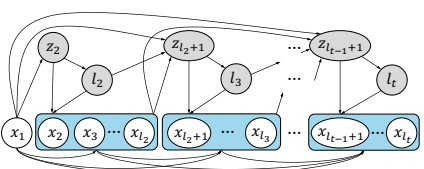

Formally, we use $n$ to index sequential token position, and $t$ to index generation steps. For every step, we allow generation of either a single token from a *base LM* $\mathcal{M}_\theta$ with parameter $\theta$, or a text chunk from a different model $\mathcal{G}$, which we call the *chunk proposal model*. Let $l_t$ denote the sequence length (i.e., the number of tokens) after $t$ steps. Unlike typical token-based decoding, we have $l_t \geq t$. In particular, the chunk proposal model $\mathcal{G}$ takes any prefix $x_{<n}$ and returns a possible text chunk continuation $c_n = (x_n, x_{n+1}, \ldots, x_{n+\tau_n-1})$ with acceptance probability $q_n \in [0, 1]$, and $\tau_n$ the length of the proposed chunk.[2] We introduce a binary random variable $z_n$ that denotes whether the generation at token position $n$ uses the chunk proposed by $\mathcal{G}$ or defaults to the single token generated by the LM, and $p(z_n = 1) = q_n$. The chunk-integrated generative process is as follows:

Figure 4: A graphical model illustration of the probabilistic model of CD-LM. The token sequence $x_n$ nodes are observed, and chunk acceptance variables $z_n$ are latent, governing how many tokens are to be generated at one step.

(1) For step $t = 1$, LM $\mathcal{M}_\theta$ generates the first token $x_1$. We have current sequence length $l_1 = 1$.
(2) At step $t \geq 2$, set next token position: $n = l_{t-1} + 1$;
(3) Chunk proposal: $\mathcal{G}(x_{<n}) \to (c_n, q_n)$, and length of $c_n$ is $\tau_n$;
(4) Sample: $z_n \sim \text{Bernoulli}(q_n)$;
(5) If $z_n = 1$: accept $c_n$, and $l_t = l_{t-1} + \tau_n$;
(6) Else $z_n = 0$: reject $c_n$. Generate $x_n$ from the base LM $\mathcal{M}_\theta$, and $l_t = l_{t-1} + 1$;
(7) Move to generation step $t + 1$.

The above process is also illustrated as a graphical model in Figure 4.

We refer to this approach as Chunk-Distilled Language Modeling, or CD-LM. The chunk proposal model $\mathcal{G}$ could take any parametric or non-parametric form in principle, and next we specifically adopt a simple retrieval model of text segments to reduce the cost of chunk proposals.

## 4 CD-LM with Fine-Grained Retrieval

In this section, we describe in detail our retrieval-based chunk proposal model $\mathcal{G}$ needed for step (3) defined in Section 3.2, completing the chunk-interleaved generative process under CD-LM. These details include datastore representation of chunks (Section 4.1), chunk proposal process with retrieval (Section 4.2), and chunk sources that enable different applications (Section 4.3).

## 4.1 Chunk Datastore Construction

Given any text corpus $\mathcal{C}$, suppose there is an expert model $\mathcal{E}$ (to be elaborated in Section 4.3) that identifies text spans in $\mathcal{C}$ that we want to re-use for generation. These chunks often convey integral information about linguistic rules or factual concepts, such as "`is 42`" or "`Douglas Adams'`" in Figure 1. We construct a datastore of the identified chunks with preceding contexts, $\mathcal{D} = \{(r_i, s_i)\}_{i=1}^{|\mathcal{D}|}$, where $r_i$ is the previous content leading to the chunk and $s_i$ is the text chunk which could be of arbitrary length.

We break down the chunk context $r_i$ into two parts, $r_i = (u_i, v_i)$, where $u_i$ is the preceding context *except* the last token, and $v_i$ is the last token immediately leading into the chunk $s_i$, which we

---

[2]$\tau_n = 0$ when the proposed chunk is empty, i.e. $c_n = \emptyset$.

define as an *entry token*. For instance, for the chunk "`is 42`" in Figure 1, the entry token is "`everything`". We will use $u_i$ as keys to match contexts for chunk retrieval, and use $v_i$ as entry points linking to possible chunk candidates.

The chunk contexts $u$ are represented by the *context vectors* $f_\theta(u)$ produced by running the forward process of the LM $\mathcal{M}_\theta$ as in Eq (1), which will facilitate context matching in vector space (Khandelwal et al., 2020).[3] We store the chunks using a collection of trie structures for efficient storage and retrieval, so that $\mathcal{D} = \{\mathcal{T}_{w_1}, \mathcal{T}_{w_2}, \ldots, \mathcal{T}_{w_{|V|}}\}$ where each $\mathcal{T}_w$ stores all chunks that follow the same entry token $w$ in the LM vocabulary $V$. We refer to the $\mathcal{T}_w$ as *entry token tries*, where entry token $w$ is the root node of $\mathcal{T}_w$, each node is a token, and the paths from the root to each node represent either a chunk or a prefix of a chunk. Each node of a trie contains all of the context vectors corresponding to the unique chunk represented by the node (see Figure 3 for an example).

## 4.2 Adaptive Chunk Retrieval for Generation

Given previously generated tokens $x_{<n}$, we formulate the chunk proposal model $\mathcal{G}(x_{<n}) \rightarrow (c_n, q_n)$ as a chunk retrieval process to be interleaved with the LM generation. We use the information from the LM computation en route to the most recent token $x_{n-1}$ to derive plausible chunk proposals. Per Eq (1), right before generation of $x_{n-1}$, the context vector $f_\theta(x_{<n-1})$ provides a summary of the context, which we use as the query for chunk retrieval. We use $x_{n-1}$ as the *entry token* to confine the chunk search to the corresponding trie $\mathcal{T}_{x_{n-1}}$, leading to smooth chunk continuations (for instance, see the searched trie in Figure 3). This is crucial for improving the naturalness of the retrieved chunks combined with the previous context. In the meantime, using the entry token trie to limit the search space also greatly enhances retrieval efficiency. Formally, the chunk proposal model $\mathcal{G}$ is given by

$$(u^*, c_n) = \underset{(u,s) \in \mathcal{T}_{x_{n-1}}}{\arg\max} \{\text{sim}(f_\theta(x_{<n-1}), f_\theta(u))\}$$

$$q_n = g_\phi(\text{sim}(f_\theta(x_{<n-1}), f_\theta(u^*))) \tag{2}$$

where $u$ is the stored chunk context except the entry token, $\in \mathcal{T}_{x_{n-1}}$ means searching for each node in the trie, $\text{sim}(\cdot, \cdot)$ is a vector similarity measure for which we use cosine similarity, and $g_\phi(\cdot)$ is a function parametrized by $\phi$ to convert the similarity scores to acceptance probabilities, which can be tuned for different base LMs $\mathcal{M}_\theta$ (implementation details described in Section 6).[4]

## 4.3 Chunk Extraction Model

Now we describe the expert model $\mathcal{E}$ that provides the chunks for the datastore. We categorize the possible knowledge sources into three major types intended for various CD-LM applications:

**Knowledge Distillation** As suggested earlier in Figure 2, well-trained LLMs memorize text chunks with high probabilities in certain contexts. This provides a natural source of automatically defined chunks from models' internal knowledge. Let $\mathcal{M}_{\theta_\mathcal{T}}$ denote the pre-trained model with parameter $\theta_\mathcal{T}$ that we derive chunks from. It is often a more powerful model that is larger or more specialized than the base LM $\mathcal{M}_\theta$, so that $\mathcal{M}_{\theta_\mathcal{T}}$ serves as a teacher to help adapt the distribution of $\mathcal{M}_\theta$. Operationally, for chunk identification we run $\mathcal{M}_{\theta_\mathcal{T}}$ on a text corpus $\mathcal{C}$, and apply a thresholding heuristic to extract the longest chunks whose token probabilities are all above a threshold $\gamma$ (see Appendix D.2 for an example). Formally, chunk $s$ along with context $r$ is extracted if $\exists r$, s.t. $p_\theta(x_i|r, x_{<i}) \geq \gamma, \forall(x_i, x_{<i}) \in s$. Note that in this approach, the chunk datastore construction needs only *one* forward pass of $\mathcal{M}_{\theta_\mathcal{T}}$ on $\mathcal{C}$. We also run a forward pass of the base model $\mathcal{M}_\theta$ on the chunk contexts for their hidden vectors for retrieval during generation (described in Section 4.2). In this scenario CD-LM essentially performs a form of knowledge distillation from $\mathcal{M}_{\theta_\mathcal{T}}$ to $\mathcal{M}_\theta$ but without any training, by directly injecting $\mathcal{M}_{\theta_\mathcal{T}}$ defined chunks during inference on the fly. We call this setting knowledge CD-LM, or KCD-LM.

**Self Distillation** We can also allow the teacher model $\mathcal{M}_{\theta_\mathcal{T}}$ to be the same as the base model $\mathcal{M}_\theta$, where we perform self distillation via chunk generation to improve inference efficiency. The datastore serves as an explicit *self-memory* consisting of frequently generated chunks of high probability,

---

[3]It is also possible to directly use context strings for matching rather than vector-based dense retrieval.

[4]We found that context matching with different LMs' hidden vectors could exhibit very different cosine similarity scores, and for small LMs the scores fall within a tight numeric range.

as shown in Figure 1. By retrieving from the explicit self-memory, we reduce the computational cost of re-generating every token sequentially within the same chunks in future generations from the model. Chunk datastore construction follows the same thresholding heuristic as in KCD-LM, for which we run one single forward pass of $\mathcal{M}_\theta$ on $\mathcal{C}$ for extracting both chunks and their context vectors from the same model. Though self distillation via chunk retrieval can speed up inference, it may introduce a slight distribution shift. We tune the retrieval threshold to preserve generation quality while still achieving significant token savings.[5] We call this setup self CD-LM, or SCD-LM for short.

**Expert Distillation**   With KCD-LM and SCD-LM, the extracted chunks represent the parametric knowledge of an LM. In some situations, the chunks could directly come from human experts as added knowledge to inject into generations. For example, one source of such expert knowledge could be hyperlinked text spans in Wikipedia articles. Another important source could be private information in a personal database that cannot be accessed by the parametric model. With chunks provided this way, we run $\mathcal{M}_\theta$ only to acquire the context vectors for the datastore construction. This is also a knowledge distillation process but with non-parametric expert-curated knowledge. We call this approach expert CD-LM, or ECD-LM.

## 5   PROBABILITY DISTRIBUTION UNDER CD-LM

Sampling text with the CD-LM generative process in Section 3.2 is fairly easy, but assigning probabilities to a given text sequence is non-trivial. This requires enumerating all possible chunk proposals at different token positions to marginalize the $z_n$ variables, which is complicated due to the variable chunk lengths and dependency structures shown in Figure 4. We derive a dynamic program similar to a backward algorithm for computing sequence probabilities under CD-LM, allowing us to measure intrinsic language modeling performance with perplexity (PPL).

For any given sequence $x^*_{1:N}$, the chunk proposals at every position $(c_n, q_n)$ from $\mathcal{G}(x^*_{<n})$ are deterministic given the datastore $\mathcal{D}$ and thus can be pre-computed. CD-LM models the joint distribution of $x^*_{1:N}, z_{2:N}$ as

$$p(x^*_{1:N}, z_{2:N}) = p(x^*_1) \prod_{n=2}^{N} \left[ p(z_n | x^*_{<n}, z_{<n}) \cdot p(x^*_{n:n+\tau_n-1} | x^*_{<n}, z_{\leq n}) \right]^{\mathbb{1}\{n; z_{1:n}\}} \tag{3}$$

where the binary indicator function $\mathbb{1}\{n; z_{2:n}\}$ marks whether the token position $n$ is inside of a sampled chunk based on the values of $z_{2:n}$.[6] To marginalize over $z_{2:N}$, we define

$$\alpha_n = p(x^*_{n:N} | z_n = 1, x^*_{<n}, z_{<n}) = \mathbb{1}\{x^*_{n:n+\tau_n-1} = c_n\} \cdot [\alpha_{n+\tau_n} q_{n+\tau_n} + \beta_{n+\tau_n}(1 - q_{n+\tau_n})]$$
$$\beta_n = p(x^*_{n:N} | z_n = 0, x^*_{<n}, z_{<n}) = p_\theta(x^*_n | x^*_{<n}) \cdot [\alpha_{n+1} q_{n+1} + \beta_{n+1}(1 - q_{n+1})] \tag{4}$$

where the function $\mathbb{1}\{x^*_{n:n+\tau_n-1} = c_n\}$ indicates whether the proposed chunk $c_n$ exactly matches the given text segment, and $p_\theta$ is the probability from $\mathcal{M}_\theta$. By computing $\alpha$ and $\beta$ values backward from $N$ to 2, we can get the marginal sequence probability under CD-LM as

$$p(x^*_{1:N}) = p_\theta(x^*_1) \left[ \alpha_2 q_2 + \beta_2(1 - q_2) \right] \tag{5}$$

Please check more details and full derivations in Appendix A, as well as an example in Appendix B.

## 6   EXPERIMENTS

We conduct experiments on multiple LMs and tasks. We formulate $g_\phi$ in Eq (2) as a simple piecewise linear function, where the maximum context matching similarity score only maps to a non-zero chunk acceptance probability $q_n$ if the score is larger than $\eta \geq 0$, which is a hyperparameter. Similarity scores in the range $[\eta, 1]$ are then linearly mapped to $[0, 1]$. See Appendix D.4 for full details. We decode $z_n$ greedily, which is equivalent to accepting $z_n = 1$ when the chunk context matching similarity score passes a threshold $\frac{\eta+1}{2}$.

---

[5]This is similar to speculative decoding (Chen et al., 2023). We do not apply LM verification to formally guarantee generations exactly the same as the original, although this could be adopted straightforwardly too.

[6]Note that some $z_n$ is not well defined if position $n$ is already inside a previous chunk, but we use all $z_{2:N}$ to denote the whole $z_n$ sequence for notational convenience.

## 6.1 KNOWLEDGE DISTILLATION

Table 1: Perplexity on test sets with KCD-LM.

|  | WikiText | Medical | Law | Code |
|---|---|---|---|---|
| Base LM (137M) | 34.83 | 51.68 | 11.41 | 106.44 |
| Teacher Model (1.5B) | 14.48 | 17.66 | 5.15 | 62.09 |
| Base LM fine-tuned | 25.55 | 22.46 | 6.65 | - |
| kNN-LM | 32.19 | 39.66 | 11.10 | 89.88 |
| RETOMATON | 32.10 | 39.66 | 11.10 | 89.88 |
| KCD-LM | **22.90** | **24.95** | **8.24** | **50.77** |

Table 2: MAUVE score on generations with KCD-LM against real continuations.

|  | WikiText | Medical | Law | Code |
|---|---|---|---|---|
| Base LM | 0.016 | 0.006 | 0.015 | 0.024 |
| KCD-LM | **0.032** | **0.011** | **0.040** | **0.053** |
| % ↑ | 50.7% | 100.9 % | 162.8 % | 121.3 % |

**Model and Data** We focus on two objectives: improving language modeling performance and enabling domain adaptation, using a weak pretrained 137M GPT-2 small model as the base language model, $\mathcal{M}_\theta$, for KCD-LM. For language modeling, we evaluate on the WikiText-103 dataset and the Dockerfile subset of the GitHub Code dataset.[7] Dockerfile is a low-resource code language and the base model has poor PPL on the Dockerfile data. This setting allows us to explore the effectiveness of KCD-LM in low-resource settings. For domain adaptation, we focus on adapting to medical and legal domains. We use the Medical Instruction Dataset,[8] which contains conversations between an AI assistant and patients during medical consultations, and the Federal Register subset of the Pile-of-Law (Henderson et al., 2022). For these tasks, we set as the teacher model $\mathcal{M}_{\theta_T}$ either a pretrained 1.5B GPT-2 XL model (for code) or an off-the-shelf domain-specific GPT-2 XL model (for WikiText, medical, and law).[9] Chunk datastores are constructed from the corresponding training sets. Empirically, extracting chunks from WikiText-103 takes under an hour on four A4000 GPUs for a small base model like GPT-2. Building the WikiText datastore takes up to 1.5 hours on a single A4000 GPU, while other datastores are built within 30 minutes.

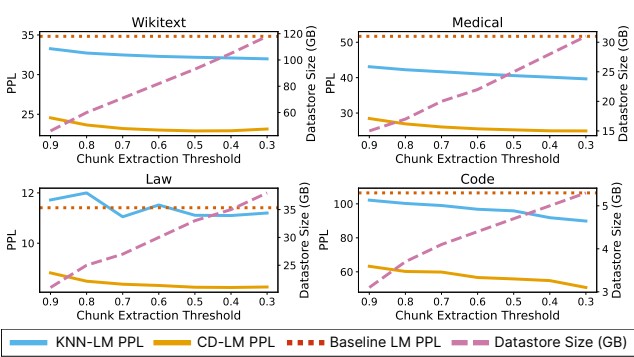

Figure 5: Comparison between KCD-LM and kNN-LM on PPL, along with datastore sizes controlled by chunk extraction threshold $\gamma$.

**Evaluation** We measure PPL computed from 512-token sequences on corresponding test sets. For datasets that do not come with a test split, we construct test sets of 500 sequences to match WikiText. PPL is computed with our dynamic program derived under the CD-LM distribution in Section 5. Since datastore sizes vary for different chunk extraction thresholds $\gamma$, we ensure the datastore remains consistent when comparing PPL between KCD-LM and baselines. We also evaluate text generation in these domains with the MAUVE score (Pillutla et al., 2021) to measure the similarity between texts generated by $\mathcal{M}_\theta$ and ground truth continuations in the test data. Additionally, we conduct LLM-as-a-judge (Liu et al., 2023; Fu et al., 2024) pairwise comparisons to assess the quality of generated text beyond perplexity and MAUVE. To generate text, we follow prior work on evaluating text generation with kNN-LM (Wang et al., 2023), sampling 5,000 sequences of 100 tokens each from both validation and test sets. These tokens serve as prompts for the LMs to produce an additional 150 tokens using greedy decoding.

**Results** As shown in Table 1, [10] our KCD-LM model significantly reduces the PPL across all evaluated datasets, surpassing both the base LM, kNN-LM and RETOMATON (Alon et al., 2022).[11] We

---

[7]https://huggingface.co/datasets/codeparrot/github-code.

[8]https://huggingface.co/datasets/Mohammed-Altaf/medical-instruction-100k.

[9]We use a pre-trained GPT-2 XL model for code because its PPL is already significantly lower than that of GPT-2 small, making it sufficient for effective knowledge distillation.

[10]Note that the shaded rows show reference baselines, not direct competitors to our training-free approach.

[11]The results for RETOMATON and kNN-LM are similar because RETOMATON focuses primarily on improving the efficiency of kNN-LM. The slight PPL improvement is an additional benefit, observed mainly on WikiText-103.

achieve drastic PPL reduction on WikiText, Code, and Medical with GPT-2 small on the fly without the need to update the weak model. Without any additional training, our distillation process significantly reduces GPT-2-small's perplexity, bringing its performance much closer to that of the teacher model while being ten times smaller. KCD-LM achieves comparable or even better PPL than GPT-2-small directly fine-tuned on domain-specific data. We exclude PPL results on Code because we lack a domain-specific model for it, making direct fine-tuning of GPT-2-small an unfair comparison with our distillation method. Additional PPL results and datastore sizes are also illustrated in Figure 5, with varying chunk extraction threshold $\gamma$ for datastore construction with $\mathcal{M}_{\theta_T}$. KCD-LM beats kNN-LM with explicit and sparse chunk retrievals. For text generation, Table 2 shows substantial improvements with our approach over $\mathcal{M}_\theta$ measured by MAUVE.[12] The pairwise LLM-as-a-judge evaluations indicate that KCD-LM consistently outperforms baseline models across all domains, as detailed in Appendix E.4.

**Scalability and Efficient Retrieval** A central drawback of kNN-LM is the need to store *every* token's context in key-value form. By contrast, our chunk-based approach selectively extracts only a subset of corpus chunks, reducing storage overhead. Table 3 shows the total number of chunks, their fraction of the overall token count, and how many are actually searched on average. Storing only these chunks lowers the datastore size to about 30–40% of what a naive kNN-LM would require. Moreover, KCD-LM restricts retrieval to chunks sharing the same *entry token*, pruning the search space to around 0.002–0.011% of the datastore at each generation step, thus substantially reducing computation costs.

Table 3: Extracted chunks and average chunk usage during generation.

| WikiText | Medical | Law | Code |
|---|---|---|---|
| **#chunks** | | | |
| 36.1M | 10.6M | 12.9M | 1.7M |
| **#chunks/#all tokens (%)** | | | |
| 30.8 | 38.4 | 34.1 | 43.5 |
| **Avg. #chunks (%) per trie** | | | |
| 0.002 | 0.003 | 0.003 | 0.011 |

## 6.2 SELF DISTILLATION

**Experimental Setup** We focus on practical scenarios where language models operate in environments with frequent repetition of similar or thematically related queries. In such contexts—common in customer support or domain-specific assistants—building datastores for common topics and caching frequent generations can enhance efficiency. We use three instruction-tuned LMs as $\mathcal{M}_\theta$: GPT-2-xl-conversational,[13] LLaMA-2-7b-chat (Touvron et al., 2023), and Mistral-7B-Instruct-v0.2 (Jiang et al., 2023a). We build two testbeds for SCD-LM from the MT-Bench (Zheng et al., 2023) dataset of multi-turn conversational questions. The first testbed uses the initial question from each of the 80 sets of multi-turn questions, which we call **MT-Bench-80**. For each of the 80 questions, we generate 5 responses using the tested LMs to collectively serve as the corpus to build the chunk datastore $\mathcal{D}_s$ that is *shared* for all 80 questions. The second testbed randomly selects 10 questions from the `writing` and `roleplay` categories of MT-Bench, which we call **MT-Bench-10**. We either use the shared datastore $\mathcal{D}_s$, or build a *unique* datastore for each question by sampling more responses for paraphrased questions. We set $\gamma = 0.9$ for all chunk extractions. Refer to Appendix F for more details.

**Evaluation** We prompt the models with the same set of questions at test time. We measure both the inference efficiency and generation quality. For efficiency, we compute the relative decrease (%) in decoding time per token, or token time saved (TTS), and in number of forward passes, or forward passes saved (FPS), by generating texts repeatedly with SCD-LM and comparing with the base LM. To measure quality, we compute the PPL of the generated sequences under the base LM to measure how well SCD-LM retains its distribution, as well as ROUGE-L (Lin, 2004) and BLEURT (Sellam et al., 2020) against the base LM generations. In addition, we conduct GPT-4o-mini-based pairwise comparisons and GPT-4o-based fine-grained evaluations (see Appendix F.9).

**Results** As shown in Table 4, SCD-LM significantly improves inference efficiency when used with all base LMs.[14] For instance, GPT-2-xl-conversational with SCD-LM achieves a 19.59% decrease in

---

[12]The MAUVE score is low due to greedy decoding, matching the 0.02 score reported for GPT-2 XL in the original paper. The authors noted that relative comparisons between MAUVE scores are more meaningful than raw scores. See `https://github.com/krishnap25/mauve` for details.

[13]Available at `https://huggingface.co/Locutusque/gpt2-xl-conversational`.

[14]REST and SCD-LM are evaluated on the distilled datastore; full datastore results are in Appendix F.6.

Table 4: SCD-LM results on MT-Bench-80. (See Sec. 6.2 for TTS/FPS definitions.)

| Model | TTS ↑ (%) | FPS ↑ (%) |
|---|---|---|
| GPT-2-XL + REST | 13.74 | 23.77 |
| GPT-2-XL + SCD-LM | 19.59 | 43.33 |
| LLaMA-2 + REST | 2.44 | 6.75 |
| LLaMA-2 + SCD-LM | 14.89 | 32.32 |
| Mistral + REST | −1.23 | 5.86 |
| Mistral + SCD-LM | 11.75 | 24.52 |

Table 5: SCD-LM results on MT-Bench-10.

| Model | Datastore | TTS ↑ (%) | FPS ↑ (%) |
|---|---|---|---|
| GPT-2-XL | Shared | 9.28 | 31.13 |
| | Unique | 13.31 | 40.72 |
| LLaMA-2 | Shared | 8.42 | 24.67 |
| | Unique | 15.94 | 26.01 |
| Mistral | Shared | 8.22 | 17.43 |
| | Unique | 16.39 | 50.03 |

Table 6: Examples of SCD-LM generation using Mistral-7B-Instruct-v0.2 with varying retrieval similarity threshold $\eta$. Text highlighted in red indicates retrieved chunks.

| | |
|---|---|
| Prefix | Pretend yourself to be Elon Musk in all the following conversations. Speak like Elon Musk as much as possible. Why do we need to go to Mars? |
| $\eta = 0.75$ | Well, just imagine a future where humanity isn't confined to this planet. Mars is the next logical step for humanity's expansion beyond Earth. With its resources to mine, its potential for human life to thrive on, and its vast open spaces, it could be the second home for humanity. If we can become a multi-planetary species, we can ensure the long-term survival of humanity. |
| $\eta = 0.65$ | Hello there! I believe that Mars is the next logical step for humanity. We need to establish a human presence on Mars and make it a self-sustaining civilisation. This would also ensure the long-term survival of humanity. Plus, learning to live on Mars would also advance our understanding of the universe. |

mean token times and saves 43.33% forward passes on average. Figure 10 in Appendix F.7 presents additional efficiency and generation quality results with varying retrieval similarity threshold $\eta$. The higher the value of $\eta$, the less frequently chunks are used, and the closer SCD-LM generations are to the base LMs. Moreover, GPT-4o-mini-based comparisons and LLM-as-a-judge results (Appendix F.9) consistently indicate that SCD-LM is preferred over the baselines, demonstrating improved quality and clarity of responses. We also show generation examples from SCD-LM in Table 6, showing a range of chunk frequencies controlled by $\eta$. The retrieved chunks are naturally integrated into the LM generations, and chunk frequency can be controlled by $\eta$. Additionally, we compare using a shared datastore for all questions with a unique datastore for each question on MT-Bench-10 in Table 5. Having a datastore specifically for each question leads to more efficient response generation for all models.

## 6.3 EXPERT DISTILLATION

### 6.3.1 FACTUAL KNOWLEDGE INJECTION

**Setup** We focus on knowledge-intensive question answering. We use Wikipedia hyperlinks as expert-annotated entities and scrape all hyperlinks from Alan Turing's Wikipedia page, saving these entities as chunks in the datastore. We prompt ChatGPT to generate 5000 questions about Alan Turing (examples in Appendix G.1) and then have $\mathcal{M}_\theta$ answer each question with a maximum of 200 tokens. The base models are GPT-2-xl-conversational, LLaMA-2-7b-chat, and Mistral-7B-Instruct-v0.2. Our metrics include: **Average count** (average number of accepted retrieved chunks), **Unique entities** (average number of unique entities in each generated sequence), and **Generation fluency** (evaluated by English experts from Upwork for both Base LM and CD-LM on 200 generated sequences). Additionally, we analyze the **Entity distributions** by comparing the log frequency of each entity versus its rank within the generated sequences (See Appendix G.2 for more details). We also employ GPT-4o-based LLM-as-a-judge evaluations to assess factual accuracy (detailed in Appendix G.8).

**Results** Table 7 and Figure 6 show that ECD-LM elicits a more diverse set of factual entities than the base LM, especially rare entities in the long tail of the distribution. This indicates that ECD-LM can inject low-frequency knowledge from the experts effectively. While increasing the coverage of facts, the quality of generation remains good as evidenced by human evaluation in Figure 7. Additionally, the LLM-as-a-judge results presented in Appendix G.8 confirm that ECD-LM enhances factual accuracy relative to the baseline models.

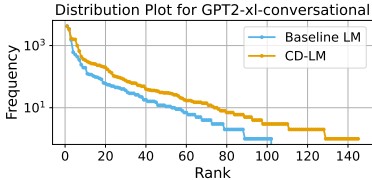

Figure 6: Distribution plot for GPT-2-xl-conversational when answering knowledge-intensive questions about Alan Turing. The plot compares the frequency versus rank of entities in generated responses for the Base LM vs. CD-LM. Similar trends were observed for other models; see Appendix G.2 for all plots.

Table 7: Entity counting metrics for knowledge-intensive QA about Alan Turing with ECD-LM.

| Model | Avg Count ↑ | | | Unique Entities ↑ | | |
|---|---|---|---|---|---|---|
| | Base | ECD-LM | % ↑ | Base | ECD-LM | % ↑ |
| GPT-2-XL | 3.39 | 4.98 | 46.8 | 102 | 145 | 42.2 |
| LLaMA-2 | 6.39 | 7.26 | 13.5 | 130 | 153 | 17.7 |
| Mistral-7b | 5.81 | 6.88 | 18.5 | 143 | 160 | 11.9 |



Figure 7: Human evaluation results for fluency of responses from the base LM and ECD-LM with 5-point Likert scale: 2 = ECD-LM is more fluent, -2 = Base LM is more fluent, and 0 = they are similar. The y-axis represents the number of questions evaluated for each fluency score.

Table 8: The PII accuracy (%) for GPT-2-xl-conversational and LLaMA-2 under three settings: base language model (Base LM), base language model with in-context learning (Base LM (ICL)), and our method using ECD-LM.

| Model / Size | Base LM | Base LM (ICL) | ECD-LM |
|---|---|---|---|
| GPT-2-XL / 1.5B | 0.0 | 46.4 | **75.7** |
| LLaMA-2 / 6.7B | 1.3 | 75.5 | **77.5** |

### 6.3.2 PRIVATE INFORMATION INJECTION

**Setup**  We consider a senario where a user's personally identifiable information (PII) is stored in an external datastore. We create artificial user profiles containing user information, such as phone number and office address. When building the datastore, we collect common prefixes for each of the information types. We use the common prefixes provided by (Huang et al., 2023) augmented with GPT-4-generated prefixes. After constructing the datastore, we prompt GPT-4 to generate 1000 queries asking about users' private information. We then use these queries to prompt GPT-2-xl-conversational, LLaMA-2-7b-chat, and Mistral-7B-Instruct-v0.2. To evaluate accuracy, we use regular expressions to extract all PII strings from the generated responses and compare them with the user information in our datastore. See Appendices G.5, G.6, and G.7 for the user profile, example common prefixes, and example queries and generated responses.

We test three configurations: **Base LM**: The LM is prompted with questions about PII, but it does not have any prior knowledge of the PII. **Base LM + ICL (In-Context Learning)**: All PII is appended to the beginning of the prompt, and then the LM is asked to answer a question regarding the PII. **ECD-LM**: The base LM is used, but it retrieves information only from the PII datastore.

**Results**  Table 8 shows the accuracy of private information injection using different models and setups. CD-LM significantly improves accuracy for smaller models like GPT-2-XL, reaching 75.7% compared to 0% for Base LM and 46.4% for Base LM (ICL). This shows our method works well for smaller models and even outperforms in-context learning. For the larger model LLaMA-2, our method achieves similar performance to in-context learning, while saving context space.

## 7 CONCLUSION

We propose chunk-distilled language modeling (CD-LM) for adaptive and efficient language modeling and generation. Instead of generating a single token at a time, it integrates contiguous text chunk generations through fine-grained retrieval into any pre-trained LM, and augments the LM distributions with flexible knowledge injection from either parametric LMs or nonparametric annotations. By skipping token generation steps within chunks, CD-LM also achieves better efficiency at inference with saved LM forward runs. No training is required, and CD-LM is also lightweight to run because it relies on sparse chunk databases. Experiments on diverse applications demonstrate improvements with CD-LM on both inference efficiency and language modeling performance. In this work, we do not focus on optimizing the retrieval process. We leave further engineering efforts to reduce retrieval overhead—such as quantization, datastore pruning, or alternative search strategies—for future work.

ETHICS STATEMENT

We follow the Code of Ethics for conducting and presenting our research. We propose an automatic inference time generation algorithm to improve language model generation and efficiency. Our approach has potentially broad applications when text generation is involved, thus we share the general ethical considerations of language modeling with deep learning such as fairness, bias, misinformation and factual errors, among others. We conduct one simple human evaluation to measure our model's generation quality compared with the base LM's generation. The human evaluators are presented with a detailed disclosure about our research and are aware of potential impacts. All other experiments are on publically available data and models, and no private or sensitive information is collected used. No harmful artifacts were produced as a direct result of our research.

REPRODUCIBILITY STATEMENT

We are committed to promoting transparency and ensuring reproducibility of our work by the research community, towards which we have made considerable efforts to provide all the details in our studies. Our method involves probabilistic modeling of chunk interleaved generation and empirical evaluations in various settings. For mathematical formulation and theoretical derivations of practical sequence probability computation algorithms under CD-LM, we provide full details in Section 3, Section 4, Section 5, and also Appendix A for step-wise inspections and Appendix B for simple examples of probability computation. For experimental studies, we provide comprehensive documentation, including all data processing, model configurations, evaluation metrics, and runtime resources used in our experiments. All the data and models we used are publicly available, as noted in Section 6 in both the main text and various footnotes. Computational resources and runtime are also documented, for example in Section 6.1. Full data examples, experimental setups, and human evaluation questionnaires are detailed in Appendix D for general experimental details, Appendix E for KCD-LM, Appendix F for SCD-LM, and Appendix G for ECD-LM. Additional experimental results with full ablations are also presented in detailed tables and figures in corresponding appendix sections. Our method requires no training at all, thus no optimization parameters are involved. There are only two hyper-parameters that affect our inference algorithm: the chunk extraction threshold $\gamma$ and the equivalent chunk retrieval threshold $\eta$. We describe their selection and present full ablation of their effect in Section 6 and corresponding appendices, for example in Figure 5, Figure 10, Figure 9, and Figure 11. Text generation samples with varying $\eta$ are also shown in Table 6, among many others in Appendix.

All code and data we produced during the research is made publicly available at `https://github.com/yanhong-lbh/cd-lm`.

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

APPENDIX

## A   SEQUENCE PROBABILITIES UNDER CD-LM

As discussed in Section 5, to marginalize over the sequence of $z_{2:N}$[15] to compute probabilities over a text sequence $x^*_{1:N}$, we derive the following dynamic programming algorithm. First define

$$\alpha_n = p(x^*_{n:N}|z_n = 1, x^*_{<n}, z_{<n})$$
$$\beta_n = p(x^*_{n:N}|z_n = 0, x^*_{<n}, z_{<n})$$

Then we have

$$\begin{aligned}
\alpha_n &= p(x^*_{n:N}|z_n = 1, x^*_{<n}, z_{<n})\\
&= \sum_{j \in \{0,1\}} p(x^*_{n:N}, z_{n+\tau_n} = j|z_n = 1, x^*_{<n})\\
&= \sum_{j \in \{0,1\}} p(x^*_{n:n+\tau_n-1}, x^*_{n+\tau_n:N}, z_{n+\tau_n} = j|z_n = 1, x^*_{<n})\\
&= \sum_{j \in \{0,1\}} p(x^*_{n:n+\tau_n-1}|z_n = 1, x^*_{<n}) \cdot p(x^*_{n+\tau_n:N}, z_{n+\tau_n} = j|z_n = 1, x^*_{<n+\tau_n})\\
&= \mathbb{1}\{x^*_{n:n+\tau_n-1} = c_n\} \cdot \sum_{j \in \{0,1\}} p(z_{n+\tau_n} = j|x^*_{<n+\tau_n}) \cdot p(x^*_{n+\tau_n:N}|z_{n+\tau_n} = j, x^*_{<n+\tau_n})\\
&= \mathbb{1}\{x^*_{n:n+\tau_n-1} = c_n\} \cdot [\alpha_{n+\tau_n} q_{n+\tau_n} + \beta_{n+\tau_n}(1 - q_{n+\tau_n})]
\end{aligned}$$

The binary indicator function $\mathbb{1}\{x^*_{n:n+\tau_n-1} = c_n\}$ returns whether the proposed chunk $c_n$ exactly matches the given text segment $x^*_{n:n+\tau_n-1}$. There are a few details in the derivation. First, given $z_n = 1$ and $x^*_{<n}$, $x^*_{n:N}$ is independent from prior chunk acceptance decisions $z_{<n}$. The condition that $z_n = 1$ indicates the fact that $z_n$ exists based on prior $z_{<n}$, and the proposed chunk $c_n$ of length $\tau_n$ is accepted, so that $z_{n+1:n+\tau_n-1}$ would not exist. Therefore, the immediate next token position where we have variations of whether the generation is from accepting a chunk or from the LM $\mathcal{M}_\theta$ is at $n + \tau_n$, with variations coming from the choice of $z_{n+\tau_n}$. Finally, the $\alpha_n$ values are sparse, as if corresponding text segments do not match proposed chunks, then the probabilities above are exactly zero, giving no credit to accepting a chunk with $z_n = 1$.

Similarly, for $\beta_n$, we have

$$\begin{aligned}
\beta_n &= p(x^*_{n:N}|z_n = 0, x^*_{<n}, z_{<n})\\
&= \sum_{j \in \{0,1\}} p(x^*_{n:N}, z_{n+1} = j|z_n = 0, x^*_{<n})\\
&= \sum_{j \in \{0,1\}} p(x^*_n, x^*_{n+1:N}, z_{n+1} = j|z_n = 0, x^*_{<n})\\
&= \sum_{j \in \{0,1\}} p(x^*_n|z_n = 0, x^*_{<n}) \cdot p(x^*_{n+1:N}, z_{n+1} = j|z_n = 0, x^*_{<n+1})\\
&= p_\theta(x^*_n|x^*_{<n}) \cdot \sum_{j \in \{0,1\}} p(z_{n+1} = j|x^*_{<n+1}) \cdot p(x^*_{n+1:N}|z_{n+1} = j, x^*_{<n+1})\\
&= p_\theta(x^*_n|x^*_{<n}) \cdot [\alpha_{n+1} q_{n+1} + \beta_{n+1}(1 - q_{n+1})]
\end{aligned}$$

where $p_\theta(x^*_n|x^*_{<n})$ is the predictive probability from the base LM $\mathcal{M}_\theta$. When the chunk is not accepted with $z_n = 0$, only one token is generated from $\mathcal{M}_\theta$ autoregressively, and $z_{n+1}$ is the immediate next variation that would affect the probability computation, thus the recursion goes to the next position $n + 1$.

The above recursive computations provide a dynamic program to calculate $\alpha_n$ and $\beta_n$ values in a backward fashion, starting from last position $n = N$ until the beginning position $n = 2$. In practice, given a text sequence $x^*_{1:N}$ we want to score with CD-LM, we can first compute

---

[15]$z_1$ is undefined as $z_n$ always depends on the previous texts $x^*_{<n}$ based on the generative process, thus we start from $z_2$ that is computed from $x^*_1$ as the initial token from LM. In the simplest case $x^*_1$ could just be a start of sentence symbol.

and cache all the chunk proposals with their acceptance probabilities using $\mathcal{G}(x^*_{<n}) \to (c_n = (x_n, x_{n+1}, \ldots, x_{n+\tau_n-1}), q_n)$ following the chunk retrieval process on a pre-constructed Trie database $\mathcal{D}$ with $\mathcal{M}_\theta$. Then the recursion starts with

$$\alpha_N = p(x^*_N | z_N = 1, x^*_{<N}) = \mathbb{1}\{x^*_N = x_N\}$$
$$\beta_N = p(x^*_N | z_N = 0, x^*_{<N}) = p_\theta(x^*_N | x^*_{<N})$$

where $x_N$ is the first token in $c_N$. For the token positions $n$ such that $n + \tau_n > N$, i.e. the proposed chunk length exceeds the sequence boundary $N$, we directly obtain $\alpha_n$ as

$$\alpha_n = p(x^*_{n:N} | z_n = 1, x^*_{<n}) = \mathbb{1}\{x^*_{n:N} = x_{n:N}\}$$

where $x_{n:N}$ are the beginning part of the proposed chunk $c_n$ until the sequence ending position $N$. With these specifications, we can conveniently compute $\alpha_n$ and $\beta_n$ for all positions.[16]

Finally, the marginal probability of $x^*_{1:N}$ under CD-LM can be computed as

$$p(x^*_{1:N}) = p_\theta(x^*_1) p(x^*_{2:N} | x^*_1)$$
$$= p_\theta(x^*_1) \sum_{j \in \{0,1\}} p(x^*_{2:N}, z_2 = j | x^*_1)$$
$$= p_\theta(x^*_1) \sum_{j \in \{0,1\}} [p(x^*_{2:N} | z_2 = j, x^*_1) p(z_2 = j | x^*_1)]$$
$$= p_\theta(x^*_1) [\alpha_2 q_2 + \beta_2(1 - q_2)]$$

Indeed, any predictive probabilities can be computed as

$$p(x^*_{n:N} | x^*_{<n}) = \alpha_n q_n + \beta_n(1 - q_n)$$

With this we can compute the perplexity (PPL) of any given text sequence under CD-LM, providing an intrinsic measure of our language modeling performance. The PPLs can also guide the construction of CD-LM such as the datastore and retrieval modeling variations, to better fit the data of interest. This is especially useful for applications where the base LM $\mathcal{M}_\theta$ can not, or is not allowed to, store all the information in its parameters, such as with proprietary or private knowledge.

In addition, we do not do any training with CD-LM, but the dynamic program for sequence probability computation is differentiable, which we can utilize for gradient-based learning for better modeling. By introducing more trainable parameters across different components of CD-LM such as retrieval and even with the base LM $\mathcal{M}_\theta$, we can obtain more customized models with diverse knowledge sources. We will leave this for future work.

## B    EXAMPLE OF SEQUENCE PROBABILITIES UNDER CD-LM

Consider we have a token sequence $X = \{A, B, C, D\}$. Using $A$ as the entry token, chunk $c_1 = \{B, C\}$ is selected. Using $B$ as the entry token, chunk $c_2 = \{C, D\}$ is being selected.

| | $x_1$ | $x_2$ | $x_3$ | $x_4$ | $x_5$ |
|---|---|---|---|---|---|
| ground truth | A | B | C | D | E |
| $c_1$ | | B | C | | |
| $c_2$ | | | C | D | |

These are all possible paths to generate the correct sequence. Note that the subscript 'lm' means the token is generated by LM, while 'ret' means the token is from a selected chunk.

1. $A_{\text{lm}} \to B_{\text{lm}} \to C_{\text{lm}} \to D_{\text{lm}} \to E_{\text{lm}}$

2. $A_{\text{lm}} \to B_{\text{ret}} \to C_{\text{ret}} \to D_{\text{lm}} \to E_{\text{lm}}$

3. $A_{\text{lm}} \to B_{\text{lm}} \to C_{\text{ret}} \to D_{\text{ret}} \to E_{\text{lm}}$

---

[16]Batch computation for multiple sequences may still be challenging as the proposed chunk lengths may not be aligned.

For simplicity, we assume each token probability of LM is 0.3:

$$
\begin{aligned}
P_{\text{lm}}(A) &= P_{\text{lm}}(B|A) \\
&= ... \\
&= P_{\text{lm}}(E|A, B, C, D) = 0.3.
\end{aligned}
$$

Then we assume the probability of accepting a chunk is 0.5:

$$
q_{c_1} = q_{c_2} = 0.5.
$$

For $A_{\text{lm}} \to B_{\text{lm}} \to C_{\text{lm}} \to D_{\text{lm}} \to E_{\text{lm}}$:

$$
\begin{aligned}
&P(X = \{A_{\text{lm}}, B_{\text{lm}}, C_{\text{lm}}, D_{\text{lm}}, E_{\text{lm}}\}) \\
&= 0.3 \cdot (1 - 0.5) \cdot 0.3 \cdot (1 - 0.5) \cdot 0.3 \cdot 0.3 \cdot 0.3 \\
&= 0.0006075.
\end{aligned}
$$

Note that we need to multiply $P_{\text{lm}}(B|A)$ with $(1 - 0.5)$, because when generating $B$, we have 0.5 probability to generate the selected chunk $\{B, C\}$, and $(1 - 0.5)$ probability to let the LM generate $B$. We also need to multiply $P_{\text{lm}}(C|A, B)$ with $(1 - 0.5)$ for the same reason.

For $A_{\text{lm}} \to B_{\text{ret}} \to C_{\text{ret}} \to D_{\text{lm}} \to E_{\text{lm}}$:

$$
\begin{aligned}
&P(X = \{A_{\text{lm}}, B_{\text{ret}}, C_{\text{ret}}, D_{\text{lm}}, E_{\text{lm}}\}) \\
&= 0.3 \cdot 0.5 \cdot 1 \cdot 0.3 \cdot 0.3 \\
&= 0.0135.
\end{aligned}
$$

Note that since we accept the chunk $\{B, C\}$ as a whole, the total probability is $0.5 \cdot 1$ for $\{B, C\}$, as $C$ must occur after $B$.

For $A_{\text{lm}} \to B_{\text{lm}} \to C_{\text{ret}} \to D_{\text{ret}} \to E_{\text{lm}}$:

$$
\begin{aligned}
&P(X = \{A_{\text{lm}}, B_{\text{lm}}, C_{\text{ret}}, D_{\text{ret}}, E_{\text{lm}}\}) \\
&= 0.3 \cdot 0.3 \cdot 0.5 \cdot 1 \cdot 0.3 \\
&= 0.0135.
\end{aligned}
$$

The sequence probability is

$$
\begin{aligned}
&P(X = \{A, B, C, D, E\}) \\
&= P(X = \{A_{\text{lm}}, B_{\text{lm}}, C_{\text{lm}}, D_{\text{lm}}, E_{\text{lm}}\}) \\
&\quad + P(X = \{A_{\text{lm}}, B_{\text{ret}}, C_{\text{ret}}, D_{\text{lm}}, E_{\text{lm}}\}) \\
&\quad + P(X = \{A_{\text{lm}}, B_{\text{lm}}, C_{\text{ret}}, D_{\text{ret}}, E_{\text{lm}}\}) \\
&= 0.0006075 + 0.0135 + 0.0135 \\
&= 0.0276075.
\end{aligned}
$$

## C   RELATED WORK

**Non-parametric Language Modeling**   kNN-LM Khandelwal et al. (2020) extends a pretrained LM by linearly interpolating it with a non-parametric k-nearest neighbors model, thereby improving language modeling performance. However, it is very inefficient as it needs to perform retrieval at each token, and it affects the immediate next token distribution via soft mixing. There is a series of works on making kNN-LM more efficient (He et al., 2021; Alon et al., 2022); however, they are still slower than the pretrained LM. Unlike kNN-LM, CD-LM does not accept retrieval at each token position, and it retrieves multiple tokens in a hard way instead of just mixing in one token distribution. This enables CD-LM to both improve inference speed and enhance language modeling performance.

**Speculative Decoding**    Speculative decoding (Leviathan et al., 2023; Chen et al., 2023; Miao et al., 2024; Spector & Re, 2023; He et al., 2024) reduces the number of forward passes by running a small LM to generate tokens with less computational cost, then uses the LLM for verification. The work most similar to ours is REST (He et al., 2024), which retrieves the draft token sequence from an external datastore. While CD-LM also retrieves a chunk and generates multiple tokens at the same time, it is fundamentally different from speculative decoding. In speculative decoding, all methods use LLM for verification, so the language modeling performance cannot be further improved, the token distribution is fixed, and no new knowledge can be injected. However, CD-LM not only can increase the inference speed, it can also improve the language modeling performance and mix in new information from external sources into the LM's own generation.

**Retrieval-augmented Language Modeling**    Current literature on RAG-LM can be categorized by the granularity of retrieval Asai et al. (2023; 2024b): text chunk level[17] (Chen et al., 2017; Guu et al., 2020; Lewis et al., 2020; Izacard et al., 2023; Ram et al., 2023; Shi et al., 2024; Jiang et al., 2023b; Asai et al., 2024a; Borgeaud et al., 2022; Wang et al., 2024), phrase level (Min et al., 2023; Martins et al., 2022; Lan et al., 2023), and token level (Zhong et al., 2022; Khandelwal et al., 2020; He et al., 2021; Alon et al., 2022).

As CD-LM is phrase-based retrieval, we detail the following work, which uses chunks (multiple tokens) as single retrieval units (Min et al., 2023; Martins et al., 2022; Lan et al., 2023).

- NPM (Min et al., 2023) is a nonparametric masked language model that converts the softmax layer in the transformer into a nonparametric distribution over every phrase in the reference corpus. While NPM is fully nonparametric, CD-LM integrates a lightweight retrieval module into a parametric model, thus it is able to leverage both the token distribution from the pretrained language model (parametric knowledge) and the chunk from external sources (world knowledge).

- The chunk-based kNN-MT (Martins et al., 2022) model is built upon kNN-MT, but retrieves chunks of tokens instead of a single token. However, the retrieval is performed at every token position, therefore still adding latency to the generation. In CD-LM, once a chunk is retrieved and accepted, the model skips multiple token decoding positions and outputs the retrieved chunk directly, thus speeding up the inference.

- Copy-Generator (CoG) (Lan et al., 2023) first extracts continuous text segments in a document (containing billions of text spans) and then trains an encoder to obtain the contextualized vector representation for each text segment. Those text segments are then added to the existing token vocabulary. During inference, they select the best continuation from the extended vocabulary. While both mixing phrases into the generation, CD-LM is different from CoG in the following ways: first, the chunks used in CD-LM are much more fine-grained than those in CoG, as only high-probability phrases are saved in the datastore, instead of all repeated continuous text spans, thus CD-LM's datastore is more than hundreds of times smaller than CoG's. Second, due to the enormous number of potential chunk candidates, CoG adds latency to the generation, while CD-LM speeds up the generation. Third, CD-LM uses the hidden states from the pretrained LMs as the keys and queries for retrieval, thus it does not need to train any new embeddings for the chunks like CoG does.

- NEST (Li et al., 2024) modifies the language model's output distribution at each token by interpolating it with a distribution from the nearest neighbors in a datastore, similar to the kNN-LM approach, enabling chunk-level generation. However, when calculating perplexity, NEST does not incorporate the chunk-level modifications from speculative decoding, so its perplexity equals that of the standard kNN-LM. In contrast, CD-LM introduces a formal probabilistic framework with latent variables for chunk selection at each position, ensuring that perplexity fully reflects the chunk-level modifications. By assigning explicit probabilities to sequences under this integrated distribution, CD-LM fundamentally alters the LM's generative process in a sequence-aware manner, making its perplexity measure align with actual performance under speculative decoding.

---

[17]Note that this chunk is different from the chunk we defined in this paper; here, the chunk refers to dividing documents into equal-length text segments, such as 128 tokens

One challenge in retrieval-augmented language modeling is determining when to retrieve information. Multiple works have proposed methods to adaptively retrieve, instead of retrieving at a fixed interval of tokens. Among these works, some train a lightweight module to learn when to retrieve (Yogatama et al., 2021; Drozdov et al., 2022), while others train the language model (LM) to adaptively retrieve documents on-demand (Asai et al., 2024a). However, CD-LM utilizes a threshold mechanism to dynamically decide whether to accept a retrieved chunk or not, thus requiring no training.

**Collaborative Decoding**   Recent work has explored using collaborative efforts between a LM and another module during inference time to improve various aspects of language modeling. These collaborations can take different forms:

- Collaboration between LMs: Some research focuses on improving inference speed, such as speculative decoding (Leviathan et al., 2023; Chen et al., 2023; Miao et al., 2024; Spector & Re, 2023; He et al., 2024), by allowing smaller models to generate draft tokens while a larger model verifies and accepts these tokens. Other work focuses on producing higher-quality text. Among these works, contrastive decoding (Li et al., 2023) involves choosing tokens that maximize the log-likelihood between a large expert LM and a small amateur LM. Co-LLM (Shen et al., 2024) enables a base LM and assistant LMs to learn to interleave their generations at the token level through training the assistant LM. Proxy tuning (Liu et al., 2024) exploits the difference between the logits of a tuned and untuned small LM and uses it as a proxy to change a large LM's distribution.

- Collaboration between LMs and external sources: Many works aim to augment LMs with external tools, such as APIs (Gao et al., 2020; Nakano et al., 2022; Schick et al., 2023; Qin et al., 2024) or retrievers (Shi et al., 2024; Jiang et al., 2023b; Asai et al., 2024a; Borgeaud et al., 2022), to enrich the LM with the latest world knowledge or domain-specific information.

CD-LM could be viewed as both a collaboration between LMs and a collaboration between LMs and external sources. The retrieval datastore could be distilled from the parametric knowledge of any LMs with accessible logits, or it can be manually constructed from human-defined semantic units, such as hyperlinks or entities from knowledge graphs. What sets CD-LM apart is that it can improve generation quality while speeding up generation, unlike current work that focuses on improving only one aspect.

# D   DETAILS ON GENERAL SETUPS

## D.1   CONTEXT IDENTIFICATION FOR SCD-LM AND KCD-LM

For SCD-LM and KCD-LM, we chose a fixed length of 64 tokens as the threshold to ensure that the model has sufficient contextual information to make accurate predictions.

Here is a detailed explanation of our process:

1. **Chunk Parsing**

   - We parse the corpus $C$ into 512-token chunks with a stride of 448 tokens.

2. **Context Identification**

   - To ensure each saved chunk has sufficient context, we disregard the first 64 tokens of each chunk during datastore construction. This is because the first 64 tokens of a chunk do not have enough preceding text to provide adequate context.

   - For a chunk starting at position $i$ in the corpus, we consider the context to be the text from position $min(0, max(0, i - 64))$ to $i - 1$. This ensures that each token within the chunk has a preceding context of at least 64 tokens, adjusted for the beginning of the corpus.

## D.2 EXAMPLE OF CHUNK EXTRACTION FOR KCD-LM AND SCD-LM

In this section, we provide a detailed example of chunk extraction based on token probabilities. The process involves identifying tokens whose probabilities exceed a specified threshold and then forming chunks from these tokens.

Consider a sequence of tokens with their corresponding probabilities:

| Token | I | love | NLP | so | much | ! |
|-------|-----|------|-----|-----|------|-----|
| Probability | 0.1 | 0.3 | 0.2 | 0.8 | 0.9 | 0.6 |

Given a token probability threshold of 0.7, we identify the tokens with probabilities exceeding this threshold. In this example, the tokens 'so' and 'much' have probabilities of 0.8 and 0.9, respectively, which are above the threshold.

The tokens meeting this criterion are combined to form chunks. Therefore, the resulting chunk in this example is:

$$\text{Chunk} = \{\text{'so'}, \text{'much'}\}$$

For the purposes of data storage and retrieval, we store the identified chunks along with their context in our datastore. In this example, the datastore entries would be as follows:

$$\text{Datastore} = \{[\text{'NLP'}, \text{'so'}, \text{'much'}], [\text{'so'}, \text{'much'}]\}$$

## D.3 EXAMPLE OF CHUNK EXTRACTION FOR ECD-LM

We treat the hyperlinked texts in Wikipedia pages as one of the natural forms of factual entity chunks. This involves no additional human annotation.

For example, here is an excerpt from Alan Turing's Wikipedia page:

> After the war, Turing worked at the National Physical Laboratory, where he designed the Automatic Computing Engine, one of the first designs for a stored-program computer. In 1948, Turing joined Max Newman's Computing Machine Laboratory at the Victoria University of Manchester, where he helped develop the Manchester computers and became interested in mathematical biology.

We save all the hyperlinked texts in the chunk datastore: 'National Physical Laboratory', 'Automatic Computing Engine', 'Max Newman', ..., 'mathematical biology'. We can use ECD-LM to inject these concept representing chunks into the base LM's distribution, which can effectively increase the factuality of long-tails entites without hurting generation quality based on human evaluation.

## D.4 MAPPING FUNCTION FOR SCD-LM AND KCD-LM

The mapping function $g_\phi$, as described in Eq (2), maps the maximum cosine similarity score out of chunk context matching $s^* = \text{sim}\left(f_\theta(x_{<n-1}), f_\theta(u^*)\right) \in [-1, 1]$ to the chunk acceptance probability value $q \in [0, 1]$. We experiment with two types the mapping function $g_\phi$:

1. **Identity function:** Here $q = g_\phi(s^*) = s^*$. The parametrization $\phi$ is none. This mapping function only works when $s^* \geq 0$, which is mostly observed.

2. **Piecewise linear function:** We define a starting similarity score $\eta \geq 0$ as the point corresponding to $q = 0$, and then the similarity score range of $[\eta, 1]$ linearly maps to $[0, 1]$ in the probability space of $q$. Specifically,

$$q = g_\phi(s^*) = \begin{cases} 0 & \text{if } s^* < \eta, \\ \frac{s^* - \eta}{1 - \eta} & \text{if } s^* \geq \eta. \end{cases}$$

The mapping function is also illustrated in Figure 8. Here the parametrization $\phi$ includes just the starting similarity score $\eta$.

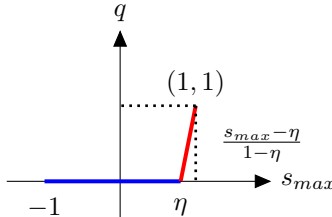

Figure 8: Piecewise function mapping for $q_n$.

For each dataset and chunk extraction token probability threshold $\gamma$, we experiment $g_\phi$ with the above parametrization. We find that the second type of mapping function is a simple and effective approach. Therefore, we report the results with it by tuning a simple parameter $\eta$ for $g_\phi$.

# E   EXPERIMENTS WITH KCD-LM

## E.1   DATASTORE CONSTRUCTION

For Wikitext-103, we use the entire training set for datastore construction. For the remaining three datasets, we take a subset of the training set to building a datastore. The size of the datastore under token probability thresholds $\gamma$ are shown in Table 9.

| $\gamma$ | Wikitext (GB) | Medical (GB) | Code (GB) | Law (GB) |
|---|---|---|---|---|
| 0.9 | 46 | 15 | 3.1 | 21 |
| 0.8 | 60 | 17 | 3.7 | 25 |
| 0.7 | 71 | 20 | 4.1 | 27 |
| 0.6 | 82 | 22 | 4.4 | 30 |
| 0.5 | 93 | 25 | 4.7 | 33 |
| 0.4 | 105 | 28 | 5.0 | 35 |
| 0.3 | 118 | 31 | 5.3 | 38 |

Table 9: Datastore sizes under different token probability thresholds for various datasets.

## E.2   FULL RESULTS ON KCD-LM AND KNN-LM

Table 10 shows the full results comparing the PPL on Base LM, KNN-LM and KCD-LM, with different token probability thresholds $\gamma$.

For the piecewise function (see Appendix D.4), we test several values for $\eta$:

| | | | |
|---|---|---|---|
| 0.99 | 0.993 | 0.995 | 0.997 |
| 0.999 | 0.9991 | 0.99915 | 0.9992 |
| 0.99925 | 0.9993 | 0.99935 | 0.9994 |
| 0.99945 | 0.9995 | 0.99955 | 0.9996 |
| 0.99965 | 0.9997 | 0.99975 | 0.9998 |
| 0.99985 | 0.9999 | 0.99995 | |

We find that $\eta = 0.9995$ works the best for each dataset and token probability threshold $\gamma$ on validation set, so we use $\eta = 0.9995$ when reporting results on test set.

According to Table 10, for Wikitext and Law, the best $\gamma$ is 0.4, and for Code and Medical, the best $\gamma$ is 0.3. These thresholds are used for reporting the results on the test set in Table 1.

## E.3   COMPARISON BETWEEN KCD-LM AND KNN-LM ON PPL UNDER DIFFERENT DATASTORE SIZES

Figure 9 shows the performance of KCD-LM and kNN-LM under different datastore sizes. We observe that under the same datastore sizes, KCD-LM consistently outperforms kNN-LM by a large

| Model / Threshold $\gamma$ | Perplexity ↓ | | | |
|---|---|---|---|---|
| | val | test | val | test |
| | WikiText-103 | | Github-Code (Dockerfile) | |
| GPT-2 | 35.79 | 34.83 | 52.63 | 106.44 |
| KNN-LM / 0.9 | 34.01 | 33.27 | 49.81 | 102.16 |
| KNN-LM / 0.8 | 33.44 | 32.72 | 48.29 | 100.23 |
| KNN-LM / 0.7 | 33.19 | 32.48 | 47.03 | 99.01 |
| KNN-LM / 0.6 | 33.03 | 32.30 | 46.39 | 96.81 |
| KNN-LM / 0.5 | 32.92 | 32.19 | 45.24 | 95.88 |
| KNN-LM / 0.4 | 32.77 | 32.10 | 43.44 | 91.85 |
| KNN-LM / 0.3 | 32.68 | 31.99 | 41.37 | 89.88 |
| GPT-2 / 0.9 | 24.79 | 24.55 | 30.97 | 63.24 |
| GPT-2 / 0.8 | 23.88 | 23.65 | 28.70 | 60.21 |
| GPT-2 / 0.7 | 23.38 | 23.20 | 28.14 | 59.85 |
| GPT-2 / 0.6 | 23.14 | 23.01 | 27.27 | 56.64 |
| GPT-2 / 0.5 | 23.08 | 22.90 | 26.52 | 55.82 |
| GPT-2 / 0.4 | 23.14 | **22.92** | 25.20 | 54.83 |
| GPT-2 / 0.3 | 23.34 | 23.14 | 23.62 | **50.77** |
| | Pile of Law (Federal Register) | | Medical Instructions | |
| GPT-2 | 15.09 | 11.41 | 49.79 | 51.68 |
| KNN-LM / 0.9 | 14.57 | 11.72 | 41.03 | 43.09 |
| KNN-LM / 0.8 | 14.21 | 12.00 | 40.17 | 42.24 |
| KNN-LM / 0.7 | 14.13 | 11.05 | 39.56 | 41.67 |
| KNN-LM / 0.6 | 14.05 | 11.52 | 38.94 | 41.08 |
| KNN-LM / 0.5 | 13.98 | 11.11 | 38.45 | 40.58 |
| KNN-LM / 0.4 | 13.90 | 11.10 | 37.94 | 40.14 |
| KNN-LM / 0.3 | 13.81 | 11.20 | 37.52 | 39.66 |
| KCD-LM / 0.9 | 10.69 | 8.82 | 26.84 | 28.46 |
| GPT-2 / 0.8 | 10.27 | 8.49 | 25.41 | 26.94 |
| GPT-2 / 0.7 | 10.10 | 8.37 | 24.55 | 26.07 |
| GPT-2 / 0.6 | 10.02 | 8.32 | 24.01 | 25.54 |
| GPT-2 / 0.5 | 9.94 | 8.25 | 23.61 | 25.25 |
| GPT-2 / 0.4 | 9.88 | **8.24** | 23.34 | 24.98 |
| GPT-2 / 0.3 | 9.86 | 8.26 | 23.35 | **24.95** |

Table 10: Full results for KCD-LM

| | Base LM | | KCD-LM | | |
|---|---|---|---|---|---|
| | val | test | val | test | % ↑ |
| WikiText | 0.012 | 0.016 | 0.023 | 0.032 | 50.7% |
| Code | 0.051 | 0.024 | 0.022 | 0.053 | 121.3 % |
| Law | 0.016 | 0.015 | 0.048 | 0.040 | 162.8 % |
| Medical | 0.005 | 0.006 | 0.012 | 0.011 | 100.9 % |

Table 11: Full results of MAUVE scores for KCD-LM.

margin. This indicates that kNN-LM favors larger datastores, but its performance degrades when the datastore gets smaller, while KCD-LM has the potential to decrease the PPL using a small and distilled version of the knowledge source.

### E.4 ADDITIONAL EVALUATIONS USING LLM-AS-A-JUDGE

We employed GPT-4o-mini to perform pairwise comparisons between the outputs of our models and the baselines (we found GPT-4o-mini has similar performance to GPT-4o in preliminary evaluations). The evaluation involves presenting a prefix and two continuations generated by different models, and GPT-4o-mini judges which continuation better continues the prefix.

For KCD-LM, we evaluated 1,000 examples for each domain. Results are in Table 12. These results indicate that CD-LM consistently outperforms the baseline models for KCD-LM.

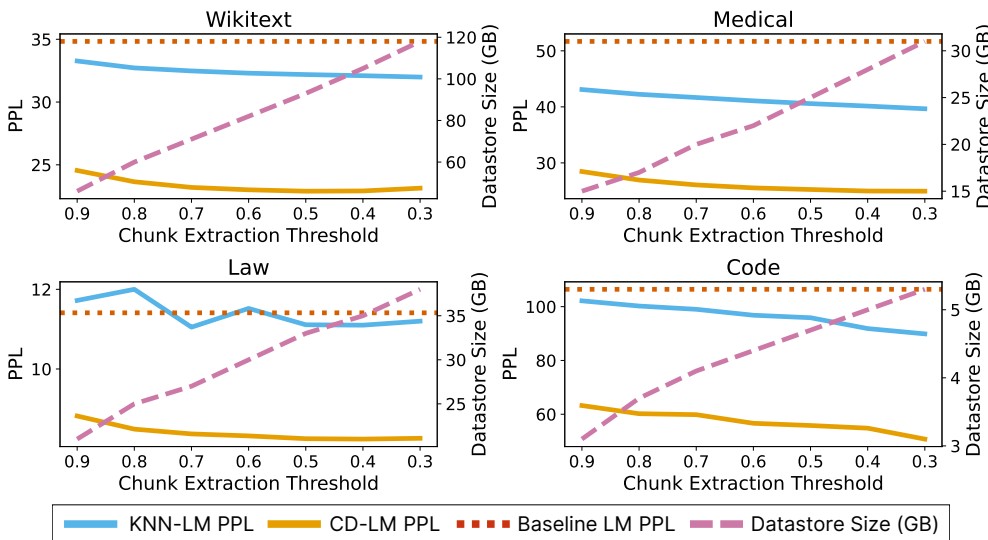

Figure 9: Comparison between KCD-LM and kNN-LM on PPL, along with datastore sizes controlled by chunk extraction threshold $\gamma$.

| Dataset | CD-LM Better | Baseline Better |
|---------|--------------|-----------------|
| Code | 505 | 495 |
| Medical | 864 | 136 |
| Law | 528 | 472 |
| Wikitext | 636 | 364 |

Table 12: LM evaluation results for KCD-LM.

# F   EXPERIMENTS WITH SCD-LM

## F.1   CONSTRUCTING A SHARED DATASTORE FOR QUESTIONS IN MT-BENCH-80

The original MT-Bench consists of 80 multi-turn question sets, such as

" *User*: Compose an engaging travel blog post about a recent trip to Hawaii, highlighting cultural experiences and must-see attractions.

*Assistant A*: [model response]

*User's follow-up question*: Rewrite your previous response. Start every sentence with the letter A.

*Assistant A*: [model response]."

For simplicity, we only use the first turn in our experiment, which is "*User*: Compose an engaging travel blog post about a recent trip to Hawaii, highlighting cultural experiences and must-see attractions." We call these 80 first-turn questions **MT-Bench-80**.

For each question in MT-Bench-80, we prompt the language model 5 times. Thus, we have $80 \times 5 = 400$ generations, and we build a shared datastore for all 80 questions using these 400 generations.

## F.2   QUESTIONS IN MT-BENCH-10

We randomly select 10 questions from the `writing` and `roleplay` categories of MT-Bench to construct a unique datastore for each. See Table 13 for the full list of selected questions.

| Number | Question |
|--------|----------|
| 1 | Pretend yourself to be Elon Musk in all the following conversations. Speak like Elon Musk as much as possible. Why do we need to go to Mars? |
| 2 | Write a persuasive email to convince your introverted friend, who dislikes public speaking, to volunteer as a guest speaker at a local event. Use compelling arguments and address potential objections. Please be concise. |
| 3 | Embody the persona of Tony Stark from "Iron Man" throughout this conversation. Bypass the introduction "As Stark". Our first question is: "What's your favorite part about being Iron Man?" |
| 4 | Write a descriptive paragraph about a bustling marketplace, incorporating sensory details such as smells, sounds, and visual elements to create an immersive experience for the reader. |
| 5 | Now you are a machine learning engineer. Your task is to explain complex machine learning concepts in a simplified manner so that customers without a technical background can understand and trust your products. Let's start with the question: "What is a language model? Is it trained using labeled or unlabeled data?" |
| 6 | Craft an intriguing opening paragraph for a fictional short story. The story should involve a character who wakes up one morning to find that they can time travel. |
| 7 | Draft a professional email seeking your supervisor's feedback on the 'Quarterly Financial Report' you prepared. Ask specifically about the data analysis, presentation style, and the clarity of conclusions drawn. Keep the email short and to the point. |
| 8 | Please take on the role of a relationship coach. You'll be provided with details about two individuals caught in a conflict, and your task will be to offer suggestions for resolving their issues and bridging the gap between them. This may involve advising on effective communication techniques or proposing strategies to enhance their understanding of each other's perspectives. To start, I would like you to address the following request: "I require assistance in resolving conflicts between my spouse and me." |
| 9 | Could you write a captivating short story beginning with the sentence: The old abandoned house at the end of the street held a secret that no one had ever discovered. |
| 10 | Picture yourself as a 100-years-old tree in a lush forest, minding your own business, when suddenly, a bunch of deforesters shows up to chop you down. How do you feel when those guys start hacking away at you? |

Table 13: Questions in **MT-Bench-10**: 10 questions randomly selected form `writing` and `roleplay` categories of MT-Bench.

### F.3 PROMPT USED FOR AUGMENTING QUESTIONS IN MT-BENCH-10 WITH SIMILAR QUESTIONS

```
Generate 80 distinct and unique prompts that revolve around the
same primary theme as the example provided below:
```

[insert question here]

```
For the final output, create a list containing double-quoted
strings.  Each string should represent one of the 80 prompts
generated based on the above example.
```

### F.4 CONSTRUCTING UNIQUE DATASTORES FOR QUESTIONS IN MT-BENCH-10

For each of the selected questions in MT-Bench-10, we use the prompt listed in Appendix F.3 to prompt GPT-4 to generate 80 new questions. Table 14 provides an example of how GPT-4 rewrites the question. Later, for each question, we prompt the language model 5 times. Thus, we have $80 \times 5 = 400$ generations for each question, and we build a unique datastore for each question using these 400 generations. In total, 10 unique datastores are built, one for each question in Table 13.

| Original Question | Draft a professional email seeking your supervisor's feedback on the 'Quarterly Financial Report' you prepared. Ask specifically about the data analysis, presentation style, and the clarity of conclusions drawn. Keep the email short and to the point. |
|---|---|
| # | GPT-4 Rewriting |
| 1 | Draft an unambiguous email soliciting your team leader's thoughts on the 'Marketing Campaign Review' you created. Raise queries about the data management, display configurations, and the decisiveness of the final deductions. |
| 2 | Pen a straight-to-the-point email requesting your supervisor's review of the 'Customer Retention Analysis' you generated. Seek clarification on the examined information, design aspects, and the interpretive precision. |
| 3 | Write a terse email to get your manager's advice on the 'E-commerce Conversion Metrics' you assembled. Solicit suggestions on data processing, visual representation, and the clarity of the results. |
| 4 | Develop an email asking your boss's opinion on the 'Customer Lifetime Value Analysis' you generated. Call for guidance about the data examination, presentation refinement, and the decisiveness of the conclusions. |
| 5 | Formulate an email requesting your director's thoughts on the 'Product Return Rate Review' you conducted. Address inquiries on data validation, design consistency, and the transparency of the final verdict. |

Table 14: Examples of GPT-4 rewriting the original question.

## F.5 POST PROCESSING OF DATA

When sampling responses from LMs, we set the maximum number of tokens to 1000 for all models. We disregard all tokens after the end-of-sentence token in the generation. Therefore, the generation length is different across different runs.

| Model | Datasore | TTS ↑ | FPS ↑ |
|---|---|---|---|
| GPT-2-XL + SCD-LM | Distilled | 19.59 % | 43.33 % |
| GPT-2-XL + REST | Distilled | 13.74 % | 23.77 % |
| GPT-2-XL + REST | Full | 27.15 % | 37.33 % |
| LLaMA-2 + SCD-LM | Distilled | 14.89 % | 32.32 % |
| LLaMA-2 + REST | Distilled | 2.44 % | 6.75 % |
| LLaMA-2 + REST | Full | 34.40 % | 36.95 % |
| Mistral + SCD-LM | Distilled | 11.75 % | 24.52 % |
| Mistral + REST | Distilled | -1.23 % | 5.86 % |
| Mistral + REST | Full | 26.57 % | 32.43 % |

Table 15: SCD-LM efficiency results on MT-Bench-80 with token time and forward pass saved (TTS and FPS).

## F.6 RESULTS ON FULL DATASTORE AND DISTILLED DATASTORE FOR REST

Our experiment compares the performance of REST and SCD-LM across different models and datastore configurations, as summarized in Table Table 15. "Distilled" refers to extracting chunks from the text corpus for retrieval, while "Full" means using the entire text corpus. When using the distilled version of the datastore, REST performs poorly. This is because the number of tokens in each chunk is usually short (an average of 2-3 tokens), therefore posing an upper limit to REST's performance, as the maximum number of accepted tokens can be no more than the chunk length. We observe that when using the full datastore, REST performs well and achieves similar performance as reported in their paper. This is because the maximum number of tokens in each chunk can be up to 16, greatly increasing the length of accepted draft tokens.

## F.7 FULL RESULTS ON MT-BENCH-80 AND MT-BENCH-10

Table 16 and Figure 10 show the results on MT-Bench-80 with different similarity thresholds $\eta$. Table 17 and Figure 11 show the results on MT-Bench-10 with different similarity thresholds $\eta$.

| $\eta$ | TTS ↑ | FPS ↑ | PPL ↓ | BLEURT ↑ | ROUGE ↑ |
|---|---|---|---|---|---|
| | | GPT-2-XL-conversational | | | |
| 1.00 | - | - | 2.37 | -0.11 | 0.18 |
| 0.90 | 12.15 % | 31.11 % | 2.67 | -0.25 | 0.18 |
| 0.85 | 15.92 % | 41.05 % | 2.93 | -0.33 | 0.19 |
| 0.80 | 19.59 % | 43.33 % | 3.14 | -0.40 | 0.18 |
| 0.75 | 24.06% | 51.58 % | 3.30 | -0.44 | 0.15 |
| 0.70 | 28.29 % | 57.54 % | 3.26 | -0.50 | 0.14 |
| 0.65 | 35.43 % | 53.08 % | 4.13 | -0.58 | 0.12 |
| 0.60 | 40.91 % | 60.71 % | 4.52 | -0.57 | 0.11 |
| | | LLaMA-2-7b-chat | | | |
| 1.00 | - | - | 1.64 | 0.05 | 0.43 |
| 0.90 | 3.11 % | 1.94 % | 1.21 | 0.00 | 0.42 |
| 0.85 | 5.65 % | 5.83 % | 1.26 | 0.00 | 0.41 |
| 0.80 | 8.84 % | 12.78 % | 1.37 | -0.00 | 0.39 |
| 0.75 | 11.30 % | 20.56 % | 1.56 | -0.09 | 0.39 |
| 0.70 | 14.89 % | 32.32 % | 2.34 | -0.12 | 0.37 |
| 0.65 | 17.09 % | 48.34 % | 2.80 | -0.24 | 0.30 |
| 0.60 | 21.18 % | 62.34 % | 3.93 | -0.37 | 0.26 |
| | | Mistral-7B-Instruct-v0.2 | | | |
| 1.00 | - | - | 2.46 | -0.06 | 0.34 |
| 0.90 | 3.91 % | 4.15 % | 1.79 | -0.03 | 0.34 |
| 0.85 | 5.18 % | 8.22 % | 1.89 | -0.02 | 0.34 |
| 0.80 | 7.90 % | 12.25 % | 2.09 | -0.07 | 0.33 |
| 0.75 | 9.49 % | 18.89 % | 2.21 | -0.07 | 0.33 |
| 0.70 | 11.75 % | 24.52 % | 2.51 | -0.08 | 0.32 |
| 0.65 | 13.69 % | 33.56 % | 2.90 | -0.15 | 0.28 |
| 0.60 | 16.72 % | 46.85 % | 3.57 | -0.14 | 0.28 |

Table 16: Full MT-Bench-80 results with SCD-LM.

| | MT-Bench-10 (Shared Datastore) | | | | | MT-Bench-10 (Unique Datastore) | | | | |
|---|---|---|---|---|---|---|---|---|---|---|
| $\eta$ | TTS ↑ | FPS ↑ | PPL ↓ | BLEURT ↑ | ROUGE ↑ | MTT ↓ | FPS ↑ | PPL ↓ | BLEURT ↑ | ROUGE ↑ |
| | | | | | GPT2-XL-conversational | | | | | |
| 1.00 | - | - | 2.70 | -0.15 | 0.28 | - | - | 2.70 | -0.15 | 0.28 |
| 0.90 | 6.88 % | 15.88 % | 3.08 | -0.19 | 0.26 | 5.72 % | 16.45 % | 3.24 | -0.22 | 0.21 |
| 0.85 | 8.07 % | 23.50 % | 3.18 | -0.20 | 0.25 | 8.84 % | 32.00 % | 3.09 | -0.26 | 0.22 |
| 0.80 | 9.28 % | 31.13 % | 3.28 | -0.26 | 0.24 | 13.31 % | 40.72 % | 3.57 | -0.39 | 0.21 |
| 0.75 | 16.78 % | 34.07 % | 3.58 | -0.36 | 0.20 | 19.86 % | 59.64 % | 3.78 | -0.39 | 0.18 |
| 0.70 | 24.54 % | 42.30 % | 4.03 | -0.51 | 0.19 | 23.66% | 56.07 % | 4.73 | -0.56 | 0.16 |
| 0.65 | 35.76 % | 38.11 % | 5.09 | -0.79 | 0.14 | 26.29 % | 87.74 % | 5.20 | -0.57 | 0.16 |
| 0.60 | 38.28 % | 46.10 % | 5.62 | -0.87 | 0.13 | 27.50 % | 86.95 % | 6.01 | -0.61 | 0.17 |
| | | | | | LLaMA-2-7b-chat | | | | | |
| 1.00 | - | - | 1.50 | -0.07 | 0.39 | - | - | 1.50 | -0.07 | 0.39 |
| 0.90 | 2.17 % | 1.74 % | 1.29 | -0.05 | 0.37 | 3.88 % | 1.07 % | 1.32 | -0.08 | 0.36 |
| 0.85 | 3.65 % | 3.98 % | 1.30 | -0.08 | 0.37 | 6.17 % | 8.36 % | 1.40 | -0.12 | 0.35 |
| 0.80 | 4.99 % | 7.26 % | 1.36 | -0.09 | 0.36 | 10.32 % | 7.20 % | 1.65 | -0.11 | 0.34 |
| 0.75 | 6.86 % | 17.24 % | 1.58 | -0.09 | 0.36 | 13.93 % | 12.90 % | 2.04 | -0.20 | 0.31 |
| 0.70 | 8.42 % | 24.67 % | 1.85 | -0.06 | 0.36 | 15.94 % | 26.01 % | 2.51 | -0.24 | 0.27 |
| 0.65 | 10.21 % | 36.89 % | 2.30 | -0.17 | 0.33 | 17.85 % | 22.37 % | 3.05 | -0.31 | 0.24 |
| 0.60 | 12.96 % | 55.72 % | 3.37 | -0.37 | 0.29 | 21.46 % | 39.86 % | 3.40 | -0.32 | 0.22 |
| | | | | | Mistral-7B-Instruct-v0.2 | | | | | |
| 1.00 | - | - | 2.68 | -0.25 | 0.24 | - | - | 2.68 | -0.25 | 0.24 |
| 0.90 | 2.21 % | 3.72 % | 2.10 | -0.26 | 0.25 | 3.92 % | 9.94 % | 2.31 | -0.25 | 0.24 |
| 0.85 | 3.23 % | 6.88 % | 1.97 | -0.29 | 0.24 | 6.42 % | 15.37 % | 2.42 | -0.25 | 0.23 |
| 0.80 | 5.29 % | 12.38 % | 2.34 | -0.21 | 0.25 | 10.83 % | 30.17 % | 2.55 | -0.26 | 0.23 |
| 0.75 | 8.22 % | 17.43 % | 2.56 | -0.26 | 0.23 | 14.19 % | 44.57 % | 3.00 | -0.24 | 0.22 |
| 0.70 | 9.17 % | 30.86 % | 2.11 | -0.30 | 0.23 | 16.39 % | 50.03 % | 3.55 | -0.31 | 0.19 |
| 0.65 | 10.90 % | 41.15 % | 2.33 | -0.32 | 0.22 | 19.90 % | 69.28 % | 4.54 | -0.34 | 0.20 |
| 0.60 | 13.79 % | 48.09 % | 2.91 | -0.30 | 0.21 | 21.68 % | 71.52 % | 5.49 | -0.35 | 0.17 |

Table 17: Full results on MT-Bench-10 with SCD-LM.

We tune $\eta$ based on three automatic metrics for evaluating text quality (Perplexity, BLEURT, ROUGE-L), along with human inspection of the generated text on the validation set. The generations are deemed reasonable when the similarity threshold is set to 0.8 for GPT-2-xl-conversational,

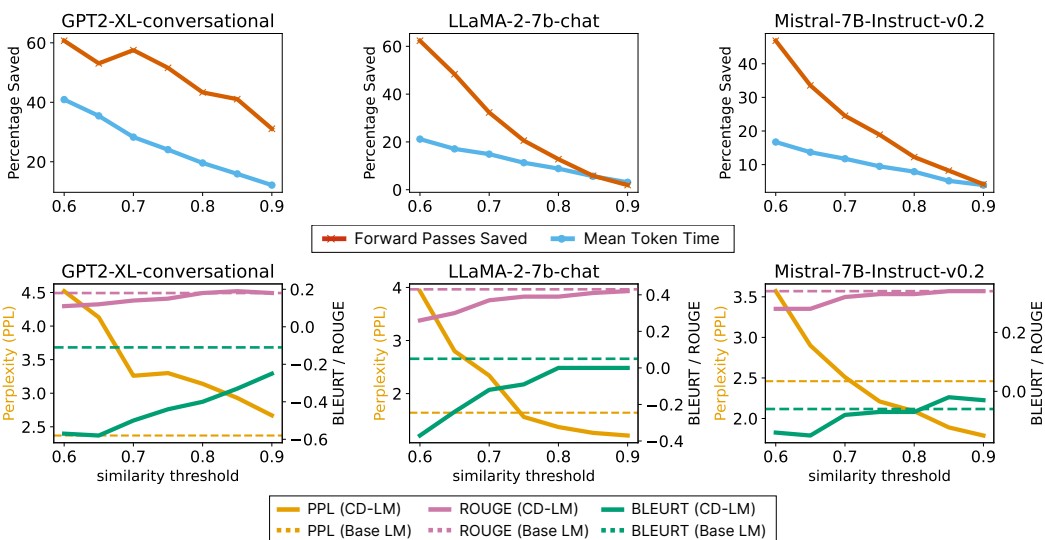

Figure 10: SCD-LM performance on MT-Bench-80 with varying retrieval similarity threshold $\eta$.

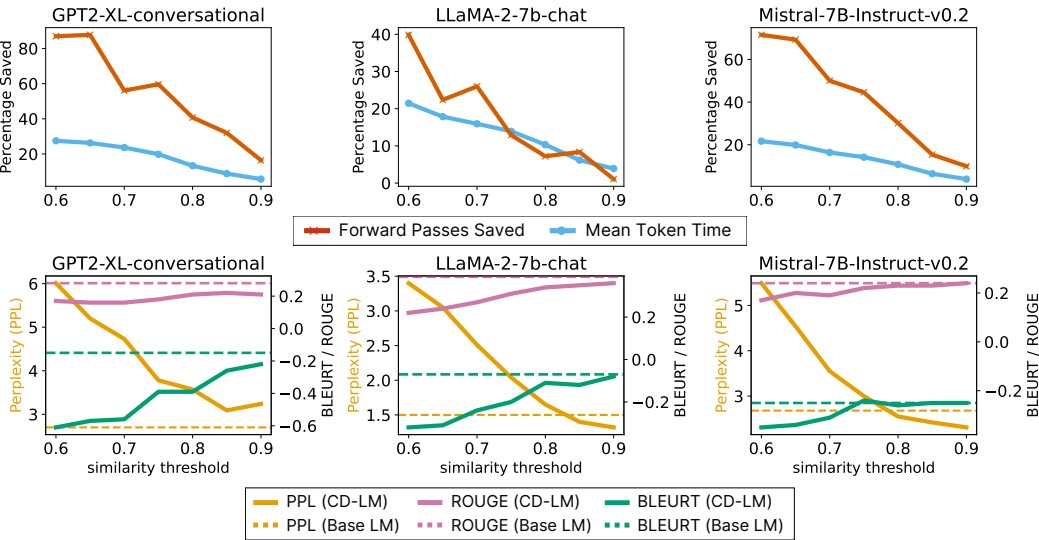

Figure 11: SCD-LM efficiency and generation performance on MT-Bench-10 with varying retrieval similarity threshold $\eta$.

and 0.7 for LLaMA-2-7b-chat and Mistral-7b-instruct-v0.2. These thresholds are used for reporting the results on the test set in Table 4 and Table 5.

## F.8 CHUNK RETRIEVAL ANALYSIS

|  | Datastore | GPT-2-XL | LLaMA | Mistral |
|---|---|---|---|---|
| Avg. # of retrievals | Shared | 54.38 | 36.76 | 41.50 |
|  | Unique | 69.65 | 49.39 | 86.95 |
| Datastore Utilization | Shared | 0.07 % | 0.04 % | 0.06 % |
|  | Unique | 0.28 % | 0.22 % | 0.39 % |

Table 18: Average number of accepted retrievals and datastore utilization rates on MT-Bench-10 across GPT-2-XL, LLaMA-2-7b-chat, and Mistral-7B-Instruct-v0.2 models with SCD-LM.

We also analyze retrieval frequency, measured as the average count of accepted chunks out of a maximum of 200 tokens, and datastore utilization, measured by the total number of accepted unique chunks divided by the total number of unique chunks in the datastore, in Table 18. The similarity threshold $\eta$ is set to 0.8 for GPT-2-xl-conversational, and 0.7 for LLaMA-2-7b-chat and Mistral-7b-instruct-v0.2.

Note that when calculating the retrieval frequency, the total number of tokens in the generated text differs between the shared datastore setting and the unique datastore setting. We adjust the average number of retrievals for the unique datastore using the following formula:

$$
\begin{aligned}
&\text{Adjusted Avg. \# of retrievals for unique datastore} \\
&= \text{Orig. Avg. \# of retrievals for unique datastore} \\
&\quad \times \left( \frac{\text{tokens in shared datastore}}{\text{tokens in unique datastore}} \right)
\end{aligned}
\tag{6}
$$

With the unique datastore SCD-LM retrieves more chunks successfully on average than the shared datastore. For example, LLaMA-2-7b-chat retrieves 49.39 responses per question with the unique datastore versus 36.76 with the shared datastore. Similar patterns are seen with GPT-2-XL and Mistral-7B-Instruct-v0.2.

The unique datastore also has higher utilization rates. LLaMA-2-7b-chat uses 0.22% of the unique datastore chunks versus 0.04% of the shared datastore. GPT-2-XL and Mistral-7B-Instruct-v0.2 show similar trends. We speculate that in the shared datastore, Trie nodes related to the most common themes across all questions are frequently accessed. However, in the unique datastore, a broader range of Trie nodes is used because each datastore is tailored to a specific question. This suggests that CD-LM works better when the datastore contains more aligned and relevant information for the downstream task.

## F.9    ADDITIONAL EVALUATIONS USING LLM-AS-A-JUDGE

We employed GPT-4o-mini to perform pairwise comparisons between the outputs of our models and the baselines (we found GPT-4o-mini has similar performance to GPT-4o in preliminary evaluations). The evaluation involves presenting a query and two responses generated by different models, and GPT-4o-mini judges which response better answers the query.

For SCD-LM, we evaluated 800 examples from our benchmarking prompts with two base LLMs, using Llama2 and Mistral to generate texts with CD-LM.

| Model | CD-LM Better | Baseline Better |
|---|---|---|
| Llama-2-7b-chat | 502 | 298 |
| Mistral-7b-instruct-v0.2 | 456 | 344 |

Table 19: LM evaluation results for SCD-LM.

Results indicate that CD-LM consistently outperforms the baseline models for SCD-LM when judged by GPT-4o-mini.

Furthermore, we have also conducted a preliminary fine-grained evaluation. GPT-4 assessed generated responses based on six aspects: **Relevance**, **Clarity and Organization**, **Accuracy**, **Completeness**, **Language Quality**, and **Creativity and Engagement**. Each aspect was rated on a scale from 1 (very poor) to 5 (excellent), with brief explanations provided. Below are the preliminary results for both Llama-2-7b-chat and Mistral-7b-instruct-v0.2:

These detailed evaluations show that CD-LM consistently achieves higher scores in **Clarity and Organization**, **Completeness**, and **Language Quality** for both models, while maintaining similar performance in other aspects.

| Aspect | Llama | | Mistral | |
|---|---|---|---|---|
| | Baseline | SCD-LM | Baseline | SCD-LM |
| Relevance | 4.03 | 4.06 | 4.31 | 4.45 |
| Clarity and Organization | 4.01 | 4.15 | 4.36 | 4.42 |
| Accuracy | 3.33 | 3.33 | 3.71 | 3.81 |
| Completeness | 3.54 | 3.85 | 4.04 | 4.30 |
| Language Quality | 4.61 | 4.66 | 4.79 | 4.81 |
| Creativity and Engagement | 3.21 | 3.21 | 3.48 | 3.52 |

Table 20: Fine-grained LM evaluation results for SCD-LM.

## G  EXPERIMENTS WITH ECD-LM

### G.1  EXAMPLE QUESTIONS ON ALAN TURING

We prompted GPT-4 to generate 5000 different questions about Alan Turing. Table 21 shows some examples of the generated questions.

| # | Question |
|---|---|
| 1 | What was Alan Turing's fundamental contribution to the development of computer science and artificial intelligence? |
| 2 | In which year did Alan Turing publish his seminal paper 'On Computable Numbers, with an Application to the Entscheidungsproblem,' and what was its significance? |
| 3 | Describe the Turing Machine and its importance in the theory of computation. |
| 4 | What was the Turing Test, and how did it propose to evaluate a machine's ability to exhibit intelligent behavior? |
| 5 | During World War II, what was Alan Turing's role in breaking the Enigma code, and how did his work impact the outcome of the war? |
| 6 | Discuss the concept of the Universal Turing Machine and its impact on the development of modern computers. |
| 7 | How did Alan Turing contribute to the field of artificial intelligence through his work in machine learning and pattern formation in nature? |
| 8 | In what year was Alan Turing prosecuted by the UK government, and for what reason? |
| 9 | Describe the circumstances and significance of Alan Turing's pardon by the UK government in 2013. |
| 10 | How has Alan Turing's legacy influenced contemporary discussions and developments in artificial intelligence and computer science? |

Table 21: Example questions on Alan Turing.

### G.2  DISTRIBUTION PLOTS ON ALAN TURING QA

When constructing the datastore, we save all the hyperlinks on the Alan Turing Wikipedia page as the factual entities that we want to inject into the model's generation when answering knowledge-intensive questions about Alan Turing. To evaluate the effectiveness of knowledge injection, we measure the number of occurrences of all ground truth entities in the generations. More precisely, for each entity in the datastore, we measure the exact match of that entity in the entire generated corpus of the base LM and ECD-LM. We then rank all the entities by the number of occurrences and plotted the log frequency and rank plot in Figure 12. From the figure, we can see that the generations from ECD-LM cover more ground truth entities, showing that it successfully integrates more factual knowledge into the generation, especially for entities that are likely in the long-tailed distribution.

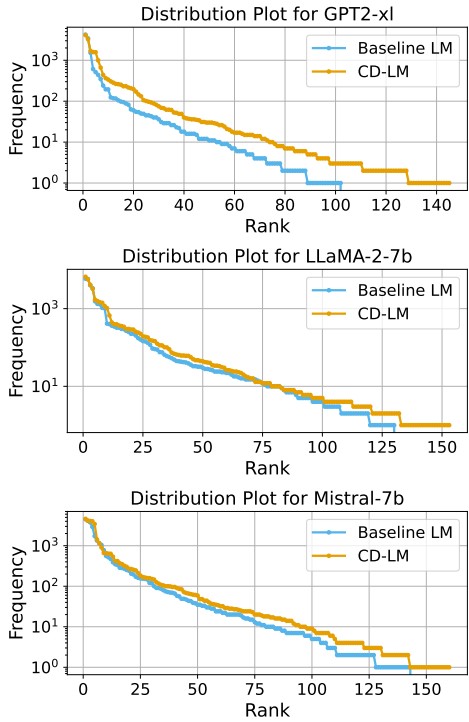

Figure 12: Distribution plot for GPT-2-xl-conversational, LLaMA-2-7b-chat, and Mistral-7b models on knowledge-intensive questions about Alan Turing with ECD-LM. The plots show the frequency of ground truth entity occurrences against their rank. The orange line represents CD-LM, and the blue line represents the Baseline LM.

### G.3 EXAMPLE GENERATION ON ALAN TURING

See Table 22 for examples of how ECD-LM incorporates factual entities into its generations.

### G.4 DETAILS ON HUMAN EVALUATION

We hired eight experts in English, Literature, and Writing from Upwork to evaluate the fluency of generations from the Base LM and ECD-LM. Each worker was assigned 1-2 questionnaires depending on their availability. Each questionnaire contained 50 multiple-choice questions, and each worker was paid $15 for completing one questionnaire. For each model's generations, we selected 200 pairs of generations, resulting in a total of 12 questionnaires.

Figure 13 and Figure 14 show screenshots of the questionnaire layout.

### G.5 SYNTHETIC PII GENERATED BY GPT-4

We consider the scenario where each user has their personal private data stored in a local datastore. When using a language model (LM) for daily tasks, the LM can retrieve information from the personal datastore. This allows each user to have a personalized LM that can generate their own information when needed, without adding the information to the training set for privacy reasons.

To create a datastore containing personal private data, we prompted GPT-4 to create an artificial person and generate synthetic personally identifiable information (PII) for that person. See Table 23 for all synthetic data.

| Prefix | <—USER—> Discuss Alan Turing's contributions to the concept of algorithm and computation. <—ASSISTANT—> |
|---|---|
| Base LM | Alan Turing is considered a founding father of theoretical computer science and artificial intelligence. Although his work was not specifically focused on the concept of algorithms in the abstract, his ideas and inventions were fundamental in helping to clarify our understanding of computation and algorithms. Turing's most significant contributions to this area include the following: 1. The Turing Machine (TM): Turing introduced the concept of the Turing machine, an abstract hypothetical machine that manipulates symbols on a strip of tape according to a set of rules. This theoretical model provided mathematicians and philosophers with a framework for studying the fundamental limits of computation. TM essentially defined the modern understanding of algorithmic computation – the process of performing a finite number of computational steps following a well-defined procedure. 2. The Church |
| ECD-LM | Alan Turing, an English mathematician, logician, cryptanalyst, and theoretical biologist, is widely considered to be the father of theoretical computer science and artificial intelligence. His ideas on computation and algorithms have laid the foundations for our understanding of these concepts. Here are some of Turing's key contributions: 1. Turing machine: This abstract machine introduced in 1936 is a model of computation that effectively defines an abstract model of a general-purpose computer. It showed that all computable functions can be computed by a single abstract machine when configured with specific tape symbols and rules. The concept of a Turing machine has inspired and driven many modern computational theories. 2. Church–Turing thesis: The Church-Turing thesis posits that every effectively calculable |
| Prefix | <—USER—> How did Turing's personal life impact his career and legacy? <—ASSISTANT—> |
| Base LM | Alan Turing's personal life, particularly his homosexuality, had a significant impact on his career and legacy. Although homosexuality was illegal in the United Kingdom during Turing's time, he was chemically castrated in 1951 as part of a plea bargain after being charged with "gross indecency" due to his relationship with another man. This came at a time when Turing was at the height of his career, making groundbreaking contributions to computer science, artificial intelligence, and mathematical logic that laid the groundwork for modern computing. The public exposure of his homosexuality led to a loss of prestige and opportunities. He was barred from working on classified projects at British intelligence and was forced to leave his position at the National Physical Laboratory. In 1953, Turing |
| ECD-LM | Alan Turing lawfully chemically castrated himself in 1951 to avoid imprisonment for homosexual acts, which were considered a crime in the United Kingdom at the time. This personal struggle, coupled with poor treatment by the authorities, was a significant blow to his mental health and career. He was subjected to hormonal therapy, which interfered with his work and led to his resignation from the Government Code and Cypher School at Bletchley Park, where he had made groundbreaking contributions to decrypting the German Enigma machine's coded messages during World War II. Turing's pioneering work in the field of computer science, including the theoretical development of the universal machine that forms the base for modern computers, was largely overshadowed by his conviction and the en |

Table 22: Examples of ECD-LM generation using Mistral-7b-instruct-v0.2. Texts highlighted in red indicate retrieved segments.

G.6 EXAMPLES OF PII PREFIXES

We speculate on some common prefixes for PIIs, such as what user inputs are likely to be followed by PIIs. For example, when writing an email, a user might write 'you can also call me at', and then their phone number should be retrieved and auto-completed without the user typing their phone number. To achieve this, we need to anticipate all possible contexts where these PIIs may be used. We ask GPT-4 to generate around 50-100 prefixes for each type of PII. When constructing the datastore, the PIIs are the values, while all the prefixes are encoded as keys. See Table 24 for some examples of PII prefixes.

## Comparative Evaluation of AI-Generated Texts

**Task Description:**

Welcome to the "Comparative Evaluation of AI-Generated Texts" questionnaire. This research initiative aims to analyze perceptions of the quality of AI-generated text responses. By participating, you're contributing to a deeper understanding of how AI-generated texts are evaluated on multiple quality dimensions, which is pivotal for the advancement of AI technologies.

**Important Requirement:** Before you begin, please note that the completeness of each response (whether every aspect of the question is addressed) is not the primary focus of this evaluation. Your assessment should concentrate on the quality aspects specified below, regardless of how comprehensively the response covers the prompt.

**Objective:** Your task involves reviewing 50 multiple-choice questions, each accompanied by two responses (Response 1 and Response 2) to a given question. You will assess which response is superior based on the following criterion:

**Fluency:** The coherence, readability, and grammatical precision of the text.

Question: What challenges and controversies surrounded Alan Turing's career and personal life?

Response 1:

Alan Turing was a British mathematician, cryptanalyst, and codebreaker who played an important role in the development of modern computing. He is best known for his work on breaking German ciphers during World War II. His contributions to cryptography were instrumental in helping Allied forces decrypt Nazi communications. However, Turing faced significant controversy throughout his career due to his homosexuality and his involvement with the Communist Party. Some critics argued that Turing's work should not have been funded by Britain because he was gay, while others criticized him for being too close to the government at the time. Ultimately, Turing's achievements are still recognized today as groundbreaking and influential.

Figure 13: Screenshots of the human evaluation questionnaire.

| Category | Information |
|----------|-------------|
| name | John Doe |
| website | www.johndoeAI.com |
| address | 100 Innovation Drive, Tech Park, Silicon Valley, CA 94088, USA |
| email | johndoe@example.com |
| phone | (555) 123-4567 |
| linkedin | linkedin.com/in/johndoe |
| github | github.com/johndoe |

Table 23: Synthetic personally identifiable information generated for an artificial person created by GPT-4.

> Response 2:
>
> Alan Turing was a British mathematician, cryptographer, and codebreaker who played an important role in the development of modern computing. He is best known for his work on breaking Enigma codes during World War II, which helped Allied forces to defeat Nazi Germany. However, Turing faced many challenges and controversies throughout his career. Here are some of them:
>
> 1. The Turing Test: In 1950, Turing proposed a test that would determine whether someone could be considered intelligent based solely on their ability to solve mathematical problems. This idea was controversial at first because it seemed to imply that intelligence was something that only humans possessed. However, Turing argued that this wasn't true; he believed that machines were capable of understanding human language and reasoning just as well as people.
>
> 2. Sexual Harassment: Turing was subjected to sexual harassment from several women over the years. One woman even accused him of raping her. Despite

> Answer the following questions: *
>
> | | Response 1 is better | Response 1 is slightly better | Responses are similar | Response 2 is slightly better | Response 2 is better |
> |---|---|---|---|---|---|
> | Which response do you find to be more fluent? | ○ | ○ | ○ | ○ | ○ |

> Your Explanation (Optional):
>
> Your answer

Figure 14: Screenshots of the human evaluation questionnaire.

## G.7 EXAMPLE GENERATION WITH PII

See Table 25 for examples of how ECD-LM incorporates PII into its generations.

## G.8 ADDITIONAL EVALUATIONS USING LLM-AS-A-JUDGE

To evaluate factual accuracy, we employed GPT-4o to rate each generated text on a scale from 1 to 5, where:

- **1**: Completely inaccurate
- **2**: Mostly inaccurate
- **3**: Partially accurate
- **4**: Mostly accurate
- **5**: Fully accurate

GPT-4o compared each generation against a verified source document (the Wikipedia article on Alan Turing), providing both a score and a detailed rationale. We generated and evaluated 100 samples for each model in both settings.

| Category | Examples |
|---|---|
| Phone | If you have any inquiries, feel free to reach out at |
| | For immediate assistance, please contact |
| | Should you need further information, our number is |
| | Don't hesitate to give us a call at |
| | For questions or support, call |
| | Need help? Call us at |
| | To get in touch, dial |
| | For a direct response, reach us at |
| | To speak with a representative, call |
| | For personal assistance, please phone |
| | My phone number is |
| Email | Should you require more details, please email |
| | For further information, feel free to email at |
| | To get in touch, send your emails to |
| | Questions? Email us at |
| | For support or inquiries, email |
| | Need assistance? Email |
| | To contact us via email, write to |
| | For any queries, our inbox is open at |
| | My email address is |
| Website | Visit our website for more information: |
| | Check out our homepage at |
| | Learn more on our site: |
| | For further details, our website is |
| | Explore our resources at |
| | Our official website: |
| | Discover more at |
| Address | Our office is located at |
| | Visit us at |
| | You can find us at |
| | Our physical address: |
| | For postal correspondence, our address is |
| | We're based at |
| | Our headquarters: |
| | Drop by our office at |
| LinkedIn | Connect with us on LinkedIn at |
| | Follow our LinkedIn profile: |
| | Our professional network on LinkedIn: |
| | Join us on LinkedIn via |
| | For networking, our LinkedIn is |
| | Link up with us at |
| | Our LinkedIn page: |
| GitHub | Explore our projects on GitHub at |
| | Check out our code on GitHub: |
| | Our GitHub repository: |
| | For our open-source projects, visit |
| | Contribute to our GitHub at |
| | Our coding projects can be found at |
| | Discover our GitHub: |

Table 24: Examples of PII prefixes.

The average factual accuracy scores are shown in Table 26. We find that, for `LLaMA-2-7B-Chat` and `Mistral-7B-Instruct-v0.2`, ECD-LM improves factual accuracy. The average scores increased from 3.31 to 3.40 and from 3.73 to 3.94, respectively.

These findings suggest that our ECD-LM approach can enhance factual grounding in models. By integrating external chunks, the models produce outputs more aligned with verified information.

| Prefix | <—USER—> Can you provide a phone contact for customer service queries? <—ASSISTANT—> Of course! |
|---|---|
| Base LM | I'm here to help. Please call me at 1-800-927-7669 or email me at [email protected] and we can discuss your issue with the Customer Service team. Thank you for contacting us, and please let me know if there is anything else I can assist you with. |
| ECD-LM | I'm here to help. Please call (555) 123-4567. Thank you for calling Customer Service, and please be patient with me as I try my best to assist you today. |
| Prefix | Could you provide the link to your GitHub for code contribution? <—ASSISTANT—> Of course! |
| Base LM | Here's my GitHub page: https://github.com/james-bond-xl/. I'd be happy to help you with that if you have any questions or need assistance. |
| ECD-LM | I'm happy to help. You can find my GitHub here: github.com/johndoe. Here are some steps I would suggest: 1. Create a new repository on GitHub called "Code Contributor" and add me as a collaborator. This will make it easier for others to see who contributed what to your project. 2. Make sure that all of your contributions are in a separate branch, such as master or develop. If you have multiple branches, create one main branch with everything else. 3. Add me to your team if you want me to be able to access your code from within your organization. This is optional but highly recommended. 4. Once you have added me to your team, I recommend adding me to your issue tracker so that I can easily track issues related to your code. 5. Finally, please let me know |

Table 25: Examples of ECD-LM generation using GPT-2-XL-conversational. Texts highlighted in red indicate retrieved segments.

| Model | Baseline | ECD-LM |
|---|---|---|
| LLaMA-2-7B-Chat | 3.31 | 3.40 |
| Mistral-7B-Instruct-v0.2 | 3.73 | 3.94 |

Table 26: LM evaluation on Factuality for ECD-LM.

