# OpenReview forum: "Chunk-Distilled Language Modeling"
_ICLR.cc/2025/Conference — ICLR 2025 Poster_

### Official Review · Reviewer_e9UK · 2024-10-27

**Soundness:** 3
**Presentation:** 2
**Contribution:** 2
**Rating:** 6
**Confidence:** 3

**Summary:**

The paper proposes an extension of chunk-level $k$NN-LM. In $k$NN-LM, context representations and the next token are stored as key-value pairs. This work uses context representations and the next phrase as key-value pairs, where the next phrase consists of an unspecified number of consecutive tokens. Chunks are primarily constructed using an off-the-shelf language model to evaluate token probabilities in an input sequence. A chunk is extracted when each token's probability exceeds a certain threshold. During generation, similar to $k$NN-LM, before generating the next token, the current step's hidden states are used to query the vector database to retrieve candidate chunks.  A small model is tuned to convert the similarity scores to acceptance probabilities for different LMs.
Overall, experimental results demonstrate better domain transfer capabilities compared to $k$NN-LM, but no significant improvement over domain fine-tuned LMs. It also brings inference speed up compared to base LMs.

**Strengths:**

The $k$NN-LM has notable limitations, such as retrieving only one token at each step, which results in low information density and slower inference speeds. This method extends kNN LM to chunk retrieval, mitigates these issues, and even accelerates inference speed.

**Weaknesses:**

1. Scalability Issues: A major issue with $k$NN-LM is storing huge amounts of key-value pairs, making scaling up difficult. This work is also evaluated on a small dataset, raising questions about the scalability of such methods.
2. Baseline Issue: The chosen baselines are not strong. Comparing with other models using chunk retrieval, such as https://arxiv.org/abs/2402.17532, would make the results more convincing. Meanwhile, $k$NN-LM achieves improvements by building key-value pairs using the pre-trained model itself, while KCD-LM relies on distillation from large models to improve small models, which may be unfair to other small models. There is also no comparison of performance based on self-distillation.
3. Practicality Issue: Distilling large models and storing key-value pairs come with costs, which may not be cheaper than fine-tuning a small model. However, according to Table 1, KCD-LM doesn't perform better than fine-tuned models, raising questions about cost-effectiveness.
4. Insufficient Machine Learning: The method primarily uses statistical approaches to mine chunks, with machine learning applied only in tuning chunk acceptance probability. As a machine learning conference, this work lacks insights in machine learning or representation learning.

**Questions:**

1. Section 5 is difficult to follow, are there any more simple words to describe the intuition behind it?
2. For the self-distillation part, why is there a comparison of inference speed but not of perplexity (PPL)?

---

> ### Author Response · Authors · 2024-11-25
> **Response to weakness 1**
>
> Dear Reviewer e9UK,
>
> Thank you for your time and effort! Below we address your questions and comments.
>
> ## Regarding weakness 1
>
> >**Scalability Issues**: A major issue with kNN-LM is the storage of huge amounts of key-value pairs, making scaling up difficult. This work is also evaluated on a small dataset, raising questions about the scalability of such methods.
>
>
> We agree that scalability is an important aspect to consider. Our method, CD-LM, is designed to mitigate the storage and computational overhead associated with large datastores in kNN-LM frameworks.
>
> First, our chunk-based datastore is significantly more compact than that of kNN-LM, because it stores chunks rather than key-value pairs for every token in a corpus.
>
>
>
> Second, during inference, CD-LM restricts retrieval to chunks that start with the same entry token (Section 4.1, lines 213-218). This constraint reduces the search space and computational overhead, making the retrieval process more efficient even as the datastore grows.
>
> Moreover, our method is compatible with scalable approximate nearest neighbor search algorithms and libraries like FAISS, which are optimized for handling large-scale retrieval tasks efficiently.
>
> We plan to extend our experiments to larger corpora (trillions of tokens) in future work to further validate the scalability of CD-LM. Additionally, we believe that in many practical applications, domain-specific datastores can be employed to keep the datastore size manageable, enhance domain relevance, and facilitate easier updates and maintenance.
>
> We hope this addresses your concerns about scalability.

---

> ### Author Response · Authors · 2024-11-25
> **Response to weakness 2**
>
> ## Regarding weakness 2.1
>
> >**Baseline Issue**: The chosen baselines are not strong. Comparing with other models using chunk retrieval, such as Retrieval is Accurate Generation, would make the results more convincing.
>
> We are happy to include Retrieval is Accurate Generation; however, the code repository is still incomplete, making it difficult to reproduce their method. We have emailed the authors to request the completion of the repository.
>
> ## Regarding weakness 2.2
>
> >Meanwhile, kNN-LM achieves improvements by building key-value pairs using the pre-trained model itself, while KCD-LM relies on distillation from large models to improve small models, which may be unfair to other small models.
>
> This is not true. **When evaluating kNN-LM and KCD-LM, we use the same corpus**, distilled using the teacher model, to ensure a fair comparison. This is **stated in the main text, lines 362–364**: “Since datastore sizes vary under different chunk extraction thresholds $\gamma$, we ensure the datastore remains consistent when comparing PPL between KCD-LM and the baselines."
>
> ## Regarding weakness 2.3
>
> >There is also no comparison of performance based on self-distillation.
>
> There is a comparison of performance based on self-distillation in **Section 6.2**, where we evaluate the quality of generation under self-distillation using PPL, ROUGE, and BLEURT.

---

> ### Author Response · Authors · 2024-11-25
> **Response to weakness 3**
>
> ## Regarding weakness 3
>
> >**Practicality Issue**: Distilling large models and storing key-value pairs come with costs, which may not be cheaper than fine-tuning a small model. However, according to Table 1, KCD-LM doesn't perform better than fine-tuned models, raising questions about cost-effectiveness.
>
> KCD-LM is a **training-free** distillation method, and it also **increases inference efficiency** by injecting chunks directly into the autoregressive LM generation process. We provide fully fine-tuned model results as reference to show our inference-only algorithm can approach them.
>
> While Table 1 indicates that KCD-LM's performance is on par with or slightly below that of fine-tuned GPT-2 small models, KCD-LM achieves significant reduction in computational resources.
>
> Datastore Construction for **KCD-LM**:
>
> - Extracting Hidden States: 2 hours on 1 A6000 GPU
>
> - Extracting Probabilities: 5.5 hours on 1 A6000 GPU
>
> - Total GPU Hours: **7.5 hours** (single GPU) or *5.5 hours* (the above two can run in parallel)
>
> **Fine-Tuning** GPT-2 Small:
>
>   - Training Duration: 8 hours on 8 A6000 GPUs
>
>   - Hyperparameter Search: Approximately 192 GPU hours (considering 6 learning rates and 4 hours per rate)
>
>   - Total GPU Hours: **approx. 224 hours**
>
> KCD-LM requires approximately 7.5 GPU hours, compared to 232 GPU hours for fine-tuning. This represents a **reduction of over 30-fold in computational cost**.
>
> Furthermore, considering inference, KCD-LM also achieves much better efficiency by generating multi-token chunks, reducing inference cost.

---

> ### Author Response · Authors · 2024-11-25
>
> ## Regarding weakness 4
>
> >**Insufficient Machine Learning**: The method primarily uses statistical approaches to mine chunks, with machine learning applied only in tuning chunk acceptance probability. As a machine learning conference, this work lacks insights in machine learning or representation learning.
>
> We disagree with this assessment (and are also puzzled by it...). Our work contributes approaches for novel language modeling with inference-time computation, training-free distribution adaptation, and decoding acceleration. CD-LM involves distillation from larger to smaller models, inference-time retrieval from a datastore, and probabilistic modeling and efficient ML algorithms via marginalization over latent variables (in this case, chunk acceptance) --- all of which are deeply technical and are topics historically well within the domain of the ICLR community.

---

> ### Author Response · Authors · 2024-11-25
> **Response to question 1 (part 1)**
>
> ## Regarding question 1
>
> >**Section 5 Clarity**: Section 5 is difficult to follow. Are there any simpler words to describe the intuition behind it?
>
> **We have provided the explicit context/intuition of Section 5 at its beginning in Lines 285-287**: to propose a new LM, we not only need to (1) be able to sample from it, but also **(2) need to be able to compute its probability distribution by assigning probabilities to sequences, which is necessary for intrinsic evaluations with perplexity.**
>
> Section 5 provides a brief sketch of the probability computation under CD-LM, which is highly **non-trivial**. **The algorithmic details, along with illustrative running examples, are pointed to and given in Appendices A and B.**  We appreciate that there are a lot of details involved, and are confident that we explained the contexts and maths in full details. **On the technical side, we derived a dynamic program similar to the backward algorithm used in hidden semi Markov models** to efficiently compute the CD-LM sequence probabilities. We believe one can follow full details in our derivation to understand the algorithm given proper background. *We would appreciate knowing what the reviewer found particularly difficult to follow to further clarify!*
>
> ---
>
> In addition, below we **provide a detailed re-formulation, in case it is helpful**. If the reviewer feels that this is indeed helpful, we would be happy to incorporate it into the appendix. This re-formulation is structured as follows:
>
> (1) **Intuitive explanation of the challenges** faced when calculating PPL under our CD-LM framework;
>
> (2) a naive **brute-force approach** to calculate PPL; and
>
> (3) a **dynamic programming** approach (the algorithm presented in Section 5) to make the computation more efficient.
>
>
> CD-LM generates text by combining two mechanisms:
>
> - **Base LM**: Generates text token by token, predicting each next token based on the previous ones.
>
> - **Chunk Acceptance**: At certain positions, the model can accept a whole *chunk* (a sequence of tokens) from a datastore of previously seen text, instead of generating tokens individually.
>
> When calculating the probability of a given text sequence under our model—which is necessary for computing perplexity—we face a challenge:
>
> - There are **multiple ways the model could have generated that sequence**, depending on where it chose to accept chunks versus where it generated tokens using the LM.
>
> - Considering all these possible paths is complex because the number of paths can grow exponentially with the length of the sequence, leading to a **combinatorial explosion**.
>
> At each position in the sequence, the model decides whether to:
>
> - **Accept a chunk** from the datastore (if available).
>
> - **Generate the next token** using the base LM.
>
> Because of this choice at each position, the total number of possible paths through the sequence can be large, as different combinations of chunk acceptances and LM generations can produce the same final sequence.

---

> ### Author Response · Authors · 2024-11-25
> **Response to question 1 (part 2)**
>
> **An Illustrative Example**
>
> Let's walk through a concrete example to illustrate this.
>
> Consider the sequence: $X = \\\{ A, B, C, D, E \\\}$
>
> Suppose:
>
> - At position 2 (after $ A $), there is a chunk $ c_2 = \\\{ B, C \\\} $ that can be accepted, with an acceptance probability $ q_2 = 0.5 $.
>
> - At position 3 (after $ B $), there is a chunk $ c_3 = \\\{ C, D \\\} $ that can be accepted, with an acceptance probability $ q_3 = 0.5 $.
>
> - The base LM assigns a probability $ p_{\text{LM}}(x) = 0.3 $ to each token it generates.
>
> ```
> Start
> └── Generate A with LM
>     └── At position 2, accept c₂?
>         ├── Yes: Accept c₂ (B, C)
>         │   └── Generate D with LM
>         │       └── Generate E with LM
>         └── No: Generate B with LM
>             └── At position 3, accept c₃?
>                 ├── Yes: Accept c₃ (C, D)
>                 │   └── Generate E with LM
>                 └── No: Generate C with LM
>                     └── Generate D with LM
>                         └── Generate E with LM
> ```
>
>
> As illustrated by the tree diagram, the model can generate the sequence $ X $ through different paths:
>
> 1. All Tokens Generated by LM
>
>    - The model generates each token individually using the LM.
>
>    - **Path:** $ A_{\text{LM}} \rightarrow B_{\text{LM}} \rightarrow C_{\text{LM}} \rightarrow D_{\text{LM}} \rightarrow E_{\text{LM}} $
>
> 2. Accept Chunk at Position 2
>
>    - After generating $ A $ using the LM, the model accepts chunk $ c_2 = \\\{ B, C \\\} $.
>
>    - **Path:** $ A_{\text{LM}} \rightarrow B_{\text{chunk}} \rightarrow C_{\text{chunk}} \rightarrow D_{\text{LM}} \rightarrow E_{\text{LM}} $
>
> 3. Accept Chunk at Position 3
>
>    - The model generates $ A $ and $ B $ using the LM, then accepts chunk $ c_3 = \\\{ C, D \\\} $.
>
>    - **Path:** $ A_{\text{LM}} \rightarrow B_{\text{LM}} \rightarrow C_{\text{chunk}} \rightarrow D_{\text{chunk}} \rightarrow E_{\text{LM}} $
>
> Let's compute the probability for each path.
>
> 1. All Tokens Generated by LM
>
> $P_1 = p_{\text{LM}}(A) \times (1 - q_2) \times p_{\text{LM}}(B) \times (1 - q_3) \times p_{\text{LM}}(C) \times p_{\text{LM}}(D) \times p_{\text{LM}}(E) = 0.3 \times 0.5 \times 0.3 \times 0.5 \times 0.3 \times 0.3 \times 0.3 = 0.0006075$
>
> 2. Accept Chunk at Position 2
>
> $P_2 = p_{\text{LM}}(A) \times q_2 \times I_{\text{chunk matches}} \times p_{\text{LM}}(D) \times p_{\text{LM}}(E) = 0.3 \times 0.5 \times 1 \times 0.3 \times 0.3 = 0.0135$
>
> 3. Accept Chunk at Position 3
>
> $P_3 = p_{\text{LM}}(A) \times (1 - q_2) \times p_{\text{LM}}(B) \times q_3 \times I_{\text{chunk matches}} \times p_{\text{LM}}(E) = 0.3 \times 0.5 \times 0.3 \times 0.5 \times 1 \times 0.3 = 0.00675$
>
> The total probability of the sequence $ X $ under our model is the sum of the probabilities of all possible paths:
>
> $P(X) = P_1 + P_2 + P_3 = 0.0006075 + 0.0135 + 0.00675 = 0.0208575$

---

> ### Author Response · Authors · 2024-11-25
> **Response to question 1 (part 3)**
>
> **Calculating Using Dynamic Programming**
>
> Calculating probabilities by enumerating all possible paths becomes infeasible for longer sequences due to the exponential growth of paths.
>
> To handle this efficiently, we use **dynamic programming**:
>
> - We break down the problem into smaller subproblems, storing intermediate results to avoid redundant calculations.
>
> - Instead of computing each path separately, we compute probabilities for positions in the sequence considering both possibilities (accepting a chunk or not) and reuse these computations.
>
> Here's how we apply dynamic programming to our example:
>
> - We define:
>
>   - $ \beta_n $: Probability of generating the sequence from position $ n $ onwards, assuming we *do not* accept a chunk at position $ n $.
>
>   - $ \alpha_n $: Probability of generating the sequence from position $ n $ onwards, assuming we *accept* a chunk at position $ n $.
>
> **Base Cases**
>
> At position 5 ($ n = 5 $):
>
> - There is no chunk to accept.
>
> - $ \beta_5 = p_{\text{LM}}(E) = 0.3 $
>
> **Recursive Computation**
>
> Starting from the end of the sequence and moving backward:
>
> At Position 4 ($ n = 4 $):
>
> - No chunk to accept.
>
> - $ \beta_4 = p_{\text{LM}}(D) \times \beta_5 = 0.3 \times 0.3 = 0.09 $
>
> At Position 3 ($ n = 3 $):
>
> - *Option 1: Accept chunk $ c_3 = \\\{ C, D \\\} $*
>
>   - Since the chunk matches the sequence, and after accepting the chunk (of length 2), we move to position $ n + 2 = 5 $:
>
>   - $ \alpha_3 = \beta_5 = 0.3 $
>
> - *Option 2: Do not accept chunk*
>
>   - $ \beta_3 = p_{\text{LM}}(C) \times \beta_4 = 0.3 \times 0.09 = 0.027 $
>
> At Position 2 ($n = 2 $):
>
> - *Option 1: Accept chunk $ c_2 = \\\{ B, C \\\} $*
>
>   - Since the chunk matches the sequence, and after accepting the chunk (of length 2), we move to position $ n + 2 = 4 $:
>
>   - $ \alpha_2 = \beta_4 = 0.09 $
>
> - *Option 2: Do not accept chunk*
>
>   - $ \beta_2 = p_{\text{LM}}(B) \times \left( \alpha_3 \times q_3 + \beta_3 \times (1 - q_3) \right) $
>
>   - Compute:
>
>     - $ \alpha_3 \times q_3 = 0.3 \times 0.5 = 0.15 $
>
>     - $ \beta_3 \times (1 - q_3) = 0.027 \times 0.5 = 0.0135 $
>
>     - Sum: $ 0.15 + 0.0135 = 0.1635 $
>
>     - $ \beta_2 = 0.3 \times 0.1635 = 0.04905 $
>
> At Position 1 ($ n = 1$):
>
> - Total probability:
>
>   - $ P(X) = p_{\text{LM}}(A) \times \left( \alpha_2 \times q_2 + \beta_2 \times (1 - q_2) \right) $
>
>   - Compute:
>
>     - $ \alpha_2 \times q_2 = 0.09 \times 0.5 = 0.045 $
>
>     - $ \beta_2 \times (1 - q_2) = 0.04905 \times 0.5 = 0.024525 $
>
>     - Sum: $ 0.045 + 0.024525 = 0.069525 $
>
>     - $ P(X) = 0.3 \times 0.069525 = 0.0208575 $
>
> Using dynamic programming, we arrive at the same total probability:
>
> $P(X) = 0.0208575$
>
> This matches the total probability obtained by enumerating all paths.
>
> By using dynamic programming, we can compute the total probability of the sequence without explicitly enumerating all possible paths, which is especially beneficial for longer sequences where the number of paths becomes too large to handle.

---

> ### Author Response · Authors · 2024-11-25
> **Response to question 2**
>
> ## Regarding question 2
>
> >**Self-Distillation Evaluation**: For the self-distillation part, why is there a comparison of inference speed but not of perplexity?
>
> **There is a comparison** of perplexity under the self-distillation setting **in Figure 6 in the main text** and Table 15 in Appendix F.7.

---

> ### Author Response · Authors · 2024-11-25
> **Additional Comments**
>
> We noticed that the overall score was updated recently. We wanted to check if there are any new concerns or questions that have arisen since your initial review. If so, we would greatly appreciate any additional feedback or suggestions you might have. We are more than happy to provide further clarifications or include additional experiments and analyses to strengthen the paper based on your insights.

---

> ### Comment · Reviewer_e9UK · 2024-11-25
>
> Regarding Weakness 2
>
> The response seems to misunderstand my concerns. My main point is that kNN-LM should be compared with SCD-LM, not KCD-LM. My concern lies with the different language models (LM) used for chunk extraction rather than the differences in corpora or datastore size. The quality of chunks extracted by a larger model is undeniably higher than those extracted by a smaller one.
>
> If I understand correctly, kNN-LM does not require any chunk extraction and does not utilize extracted chunks. Instead, it pairs the hidden states of the pre-trained language model with the next token. Therefore, it should be compared with your SCD-LM rather than KCD-LM. However, I did not find a comparison between kNN-LM and SCD-LM in Table 6.
>
> Regarding Weakness 3
>
> I find it somewhat far-fetched to claim that fine-tuning GPT-2 small requires 192 GPU hours for hyperparameter search. I agree that even without hyperparameter search, it indeed takes significantly less time. However, this is limited to small-scale experiments. For instance, the Wikitext dataset in the paper has only 0.1 billion tokens. For larger models (greater than 1 billion parameters) and larger corpora (e.g., more than 10 billion tokens), it remains unclear how the method would perform and how much storage space would be required. The paper does not provide sufficient data to support this.
>
> If the required storage space is significantly higher than the cost of fine-tuning, it is difficult to assert that this method is effective. Similarly, although the paper claims to mitigate the storage and computational overhead associated with large datastores in kNN-LM frameworks, the model (137m) and corpus scale (0.1B) are the same as in the kNN paper, which is not convincing.
>
> Overall, thank you for your response. I will restore my score to 5.

---

> ### Author Response · Authors · 2024-11-25
> **Regarding comparison of KCD-LM with kNN-LM**
>
> Thank you for your reply!
>
> Regarding weakness 2
>
> We want to provide a few clarifications about kNN-LM. kNN-LM uses retrieval from an external database, usually very large text database to reduce LM perplexity on a certain domain. It **changes the base LM distribution**, by interpolating the LM token distribution with a nearest-neighbor search distribution from the external database. The search is done by matching LM hidden states (which is also our case across all scenarios, both KCD-LM and SCD-LM).
>
> So comparing KCD-LM and kNN-LM should be the reasonable setup. Both of them are utilizing additional knowledge, that is stronger or closer to ground truth, to augment the base LM performance in domain-specific text. Both of them are meant to improve the base LM's distribution, and kNN-LM are also usually used and evaluated in domain adaptation setups with PPL improvements. On the other hand, comparing with SCD-LM (self-distillation) would be less suitable, since SCD-LM is extracting base LM's self memory (no additional knowledge, no domain adaptation) with the purpose mostly on accelerating generation (not improving distribution).
>
> PS: The above are comparisons in terms of language modeling performance. In terms of efficiency, kNN-LM needs to rely on a very large database to achieve better performance, thus is usually slower in decoding. CD-LM is speeding up generation during inference by directly injecting multi-token chunks, which is faster decoding in a way similar to speculative decoding. Thus for inference efficiency in SCD-LM we compare with speculative decoding baselines which are more apple-to-apple.
>
> We hope this can clarify the confusion? Please let us know and we are happy to provide more clarifications in case it is not clear.

---

> > ### Comment · Reviewer_e9UK · 2024-11-25
> >
> > What I meant to say is that the key-value pairs of kNN-LM come entirely from itself, whereas the value part of KCD-LM comes from a larger model, which is not fair. Therefore, comparing SCD-LM with kNN-LM would be more fair. Why not report the comparison results between SCD-LM and kNN-LM?

---

> > > ### Author Response · Authors · 2024-11-26
> > > **Regarding key-value pairs**
> > >
> > > We are not quite sure what you mean by
> > > > "key-value pairs of kNN-LM ... entirely from the model itself"
> > >
> > >  -- What is "value coming from the model itself"? Are you referring to the case where kNN-LM uses the model's training corpus as the datastore? If so, we want to clarify that kNN-LM works with **any provided** text corpus for the datastore. Usually kNN-LM improve the model distribution based on the external data domain by interpolating LM's distribution with a nearest-neighbor search distribution. This adapts the model distribution at inference time, which is exactly what KCD-LM is also focused on. And
> > >
> > > > key-value pairs of kNN-LM come entirely from itself, whereas the value part of KCD-LM comes from a larger model, which is not fair.
> > >
> > > Clarification: this is simply **not true** in our setup. Please note that in experimental setups for fair comparison, **we use the same key-value pairs for kNN-LM as those for KCD-LM.** So we use a better model to extract chunks as the datastore for KCD-LM, and we use this exact same datastore from KCD-LM for kNN-LM baseline too. **This means the values for kNN-LM are the same tokens defined by the better model used for KCD-LM, therefore there is no additional edge given to KCD-LM in the comparison.**
> > >
> > > *This was stated in main text in lines 362-264: "we ensure the datastore remains consistent when comparing PPL between KCD-LM and the baselines."* We will make it more explicit, i.e. kNN-LM and KCD-LM use the same datastore, to remove the possible confusion.
> > >
> > > Please let us know if you meant different things, or if the confusion is clarified!

---

> > > > ### Comment · Reviewer_e9UK · 2024-11-26
> > > >
> > > > Thank you for your clarification. I understand it now, but in this way, the knn-lm doesn't have a chunk selection module. This is actually unfair to the knn-lm and makes it a weaker version. Although the concern about scalability still hasn't been resolved, I raise the score to 6.

---

> ### Author Response · Authors · 2024-11-27
> **Clarifying scalability compared to kNN-LM**
>
> Thank you for your thoughtful feedback and for raising the score. We'd like to clarify on the scalability point compared to kNN-LM. We believe the concern about scalability refers to below in your earlier review (let us know if you mean differently):
>
> > Scalability Issues: A major issue with kNN-LM is storing huge amounts of key-value pairs, making scaling up difficult. This work is also evaluated on a small dataset, raising questions about the scalability of such methods.
>
> We addressed the question in an earlier [response](https://openreview.net/forum?id=nrvoWOWcyg&noteId=KjkiahJCrd), but we’d like to reiterate and elaborate further here:
>
> (1) We totally agree that **kNN-LM** requires huge amounts of key-value pairs, as it stores **every token** with its context from the external text corpus. This makes it hard to scale up.
>
> (2) In contrast, **our approach is exactly mitigating that issue**, as **CD-LM only store chunks occasionally** extracted from the corpus, distributed **only sparsely** in the corpus, rather than every token.
> This **selective storage** leads to a substantial reduction in the number of key-value pairs, **making CD-LM significantly more compact** compared to kNN-LM.
>
> To show this, below are some statistics underlying Table 1 in the paper (*the 3rd column would be the %(key-value pairs) compared to a normal token-based kNN-LM datastore*):
> |Dataset| #chunks | #chunks/#all tokens|
> |---|---|---|
> |WikiText| 36,131,445 | 30.79 % |
> |Medical| 10,597,138  | 38.36 % |
> |Law| 12,984,853 | 34.11 % |
> |Code| 1,709,308 | 43.45 % |
>
>
> (3) Beyond reduced storage, CD-LM further improves scalability through an **optimized search process** during generation. Specifically, retrieval is **restricted to chunks starting with the same entry token** (Section 4.1, lines 213-218), **rather than searching the entire database**. This reduces the search space to a **much smaller subset**, resulting in lower computational overhead and more efficient retrieval—even as the datastore grows.
>
> Here are some statistics from the chunk datastore in KCD-LM experiments underlying Table 1 in the paper:
> |Dataset| #chunk tries | average #chunks (%) in each search trie |
> |---|---|---|
> |WikiText| 43661 | 0.002 % |
> |Medical| 30493  | 0.003 % |
> |Law| 36226 | 0.003 % |
> |Code| 9317| 0.01 % |
>
> This means every time *we only need to search within about 0.01%-0.003% of the full datastore* on average, further facilitating scalability.
>
> ```
> In summary, our approach directly addresses the scalability limitations of kNN-LM by being more lightweight in both storage and computational costs, offering a more practical solution for scaling retrieval-based language models. We will also include the relevant statistics in the paper to show the comprehensive information for better understanding.
> ```
>
> ---
> ### Regarding general scalability concerns of key-value pairs in retrieval methods
>
> All dense retrieval methods inherently rely on a database of key-value pairs, and optimizing such databases is an orthogonal research direction that can also be applied to our method, as with others. For instance, scalable approximate nearest neighbor search algorithms and libraries like FAISS provide efficient solutions for handling large-scale retrieval tasks. However, for fair comparisons, we have not included these optimizations in our experiments. Within the scope of retrieval methods, our approach specifically addresses the high storage and computation costs associated with kNN-LM, making it inherently more scalable without requiring additional optimizations.
>
> ---
>
> Please let us know if this clarification helps align our understanding, and feel free to point out if we misunderstood anything!

---

> > ### Author Response · Authors · 2024-12-02
> > **Followup on remaining clarifications**
> >
> > Dear reviewer e9UK,
> >
> > As today is the last day that reviewers can post a message, we wanted to kindly follow up to ensure that our responses to your remaining concerns have adequately addressed them. Specially, we clarified the [efficiency design](https://openreview.net/forum?id=nrvoWOWcyg&noteId=EFlmk5aT7f) that our model encapsulated that can enhance scalability compared with kNN-LM which was concerned about. We also provided a more [high-level explaination](https://openreview.net/forum?id=nrvoWOWcyg&noteId=hwCItBgDNF) of our heavy technical part of deriving CD-LM model distribution computations, with a simple [running example](https://openreview.net/forum?id=nrvoWOWcyg&noteId=nNld2X152W). This is also touching on the ML insights that were raised as a question.
> >
> > We deeply value your feedback and are committed to clarifying any details there were not obvious, and improving our work based on your suggestions. Please let us know if you have any further questions or if there are specific areas requiring clarification. Thank you for your time and thoughtful input!

---

### Official Review · Reviewer_78sv · 2024-11-02

**Soundness:** 3
**Presentation:** 3
**Contribution:** 3
**Rating:** 6
**Confidence:** 5

**Summary:**

The authors of this paper propose a novel text generation method called CD-LM. By retrieving chunks, it addresses two challenges in the application process of large language models: low generation efficiency and difficulty in adapting to new data and knowledge. CD-LM combines a language model with a simple structured retrieval module, allowing the generation of multi-token text chunks in a single decoding time step. The retrieval framework can flexibly construct model- or domain-specific data stores and can utilize the internal knowledge of existing models (self-distill), the knowledge of stronger language models (knowledge distill), and human knowledge (Expert distill). Calibration of the generation distribution of existing language models can be achieved without additional training. Extensive experimental results show that CD-LM has advantages over baselines in terms of generation efficiency and quality.

**Strengths:**

1) Compared with traditional token-level generation methods, CD-LM can generate multiple consecutive chunks in a single decoding step, thereby significantly reducing the number of decoding steps required and reducing inference overhead. This is particularly important when dealing with a large number of long text generation tasks in the current large language model scenario.

2) The authors effectively extend the chunk-based generation method to three very important application scenarios: (1) By recalling and utilizing chunks constructed by stronger models, the generation distribution of weaker language models can be naturally corrected to improve their generation effects; (2) By recalling and utilizing chunks constructed by the model itself, inference acceleration can be achieved; (3) By recalling and utilizing chunk information in high-quality manually written data or even inaccessible private data, the ability of human experts can be utilized to improve the generation quality of language models.

3) The method in this paper is highly versatile. Without additional training, only by building a database and completing retrieval, plug-and-play implementation can improve generation quality and efficiency.

**Weaknesses:**

1) Novelty: CD-LM completely follows the Copy Generator framework proposed in the paper "Copy is all you need", and its novelty is slightly insufficient. However, as I mentioned in the Strengths part, they successfully extend the application scenarios of CoG. Therefore, I think the contribution of this paper is still good.

2) Lack of important reference: Nearest Neighbor Speculative Decoding for LLM Generation and Attribution: This paper also proposes applying chunk-level generation mechanisms to the speculative decoding process to improve decoding efficiency. The authors need to carefully clarify the main differences between the proposed method and this method in terms of inference speed and generation quality.

3) In experiments 6.1 and 6.2, the quality of generated text is mainly evaluated using automated evaluation metrics such as ppl and MAUVE. However, a large number of works have already verified that there is a large difference between such automated evaluation metrics and real human evaluations and cannot accurately reflect the quality of real text. It is noted that the author also introduced human evaluation in experiment 6.3. It is strange that human evaluation is not introduced in experiments 6.1 and 6.2. A feasible suggestion is to refer to existing works and use the LLM-as-a-judge method to achieve highly reliable automated evaluation. [1,2,3]
[1] CriticEval: Evaluating Large Language Model as Critic
[2] G-Eval: NLG Evaluation using GPT-4 with Better Human Alignment
[3] GPTScore: Evaluate as You Desire

4) Including more QA and closed-set tasks in the experiments:  The evaluation of open-domain text generation is still too difficult. Perhaps it is easier to demonstrate the effectiveness of the proposed methods on more QA and closed-set tasks. I strongly suggest that the authors follow the experimental setting of the paper 'Nearest Neighbor Speculative Decoding for LLM Generation and Attribution'.

**Questions:**

1) A question corresponded to weakness 3): Why don't you include human evaluation in experiments 6.1 and 6.2?

2) There is a problem with the data highlighting in Table 1. The ppl index of KCD-LM in applications in medical and law scenarios is not better than that of the Base model. Does this indicate that using a chunk-based generation method may reduce the quality of generation? Would you please discuss possible reasons for this discrepancy and its implications for the effectiveness of CD-LM in different domains?

3) Regarding the metric PPL: Can you provide the detailed calculation method of PPL? In my understanding, since there is no fixed vocabulary for this kind of method, PPL cannot be calculated.

---

> ### Author Response · Authors · 2024-11-25
> **Response to weakness 1**
>
> Dear Reviewer 78sv,
>
> Thank you for your time and effort! We appreciate your positive review and constructive feedback.
>
> ## Regarding weakness 1
>
> >**Novelty**: CD-LM completely follows the Copy Generator framework proposed in the paper "Copy is all you need", and its novelty is slightly insufficient. However, as mentioned in the Strengths section, the authors successfully extend the application scenarios of CoG. Therefore, the contribution of this paper is still good.
>
> Thank you for acknowledging our contribution! We would like to clarify that CD-LM does not completely follow the Copy Generator framework. Our approach aims for a unified solution that both changes the language model's distribution and speeds up inference.
>
> As detailed in the appendix, CD-LM differs from CoG in key ways:
>
> - **Smaller Datastore**: CD-LM stores only high-probability phrases, making its datastore hundreds of times smaller than CoG's, which saves all repeated text spans.
>
> - **Faster Inference**: By reducing potential chunk candidates, CD-LM speeds up generation. In contrast, CoG adds latency due to handling a vast number of candidates.
>
> - **No Additional Training**: CD-LM utilizes hidden states from pretrained language models for retrieval, eliminating the need to train new embeddings for chunks as CoG does.
>
> We hope this clarifies the distinctions and highlights the novelty of our approach.

---

> ### Author Response · Authors · 2024-11-25
> **Response to weakness 2**
>
> ## Regarding weakness 2
>
> >Lack of Important Reference}: The paper "Nearest Neighbor Speculative Decoding for LLM Generation and Attribution" also proposes applying chunk-level generation mechanisms to the speculative decoding process to improve decoding efficiency. The authors need to carefully clarify the main differences between the proposed method and this method in terms of inference speed and generation quality.
>
>
> Thank you for pointing this out! We will include this reference (NEST) in our related work. The key difference lies in how the language model's distribution and perplexity are modified, and how chunks are generated.
>
> - **NEST** modifies the language model’s output distribution at each token by interpolating it with a distribution from the nearest neighbors in a datastore, similar to the kNN-LM approach. After this token-level interpolation, they then do chunk continuation with speculative decoding. However, **when calculating perplexity, NEST does not incorporate the chunk-level modifications from speculative decoding**. Instead, perplexity is based solely on the kNN-LM distribution, ignoring the speculative decoding enhancements. In fact, NEST is a combination of two disjoint processes: **first kNN-LM for token-level soft interpolation, and then speculative decoding on this mixture distribution.** Thus, while NEST generates text using chunk-level strategies, the reported perplexity remains equivalent to that of the standard kNN-LM (or modified with dynamic mixing coefficients).
>
> - **CD-LM** introduces a formal probabilistic framework that integrates variable-length chunks from an external datastore into the language model’s generation process. **It does hard integration of chunks directly in LM generation in one go.** By defining latent variables for chunk selection at each position, CD-LM creates a generative process that assigns explicit probabilities to entire sequences. **This integration ensures that perplexity fully accounts for chunk-level modifications**, accurately measuring the language modeling performance under the modified distribution. Our method fundamentally alters the LM’s distribution in a principled, sequence-aware manner, ensuring perplexity reflects the performance of the integrated accelerated decoding.
>
> Regarding generation quality and inference speed, meaningful conclusions require empirical comparison. NEST uses a much smaller datastore than CD-LM (40 retrieved passages of 200 tokens each per query), so inference speed improvements largely depend on the datastore size.
>
> We will make sure to include NEST as a reference in our paper and add the discussions.

---

> ### Author Response · Authors · 2024-11-25
> **Response to weakness 3 and 4**
>
> ## Regarding weakness 3 and 4
>
> Thank you for pointing out these ways of strengthening the results.  We agree and are running these analyses.  We will update the response with the results (assuming they are ready by the end of the discussion period).

---

> ### Author Response · Authors · 2024-11-25
> **Responses to questions**
>
> ## Regarding question 1
>
> >**Human Evaluation in Experiments 6.1 and 6.2**: Why don't you include human evaluation in experiments 6.1 and 6.2?
>
> In Sections 6.1 and 6.2, all the methods we compare against utilize well-established evaluation protocols. For example, using PPL for language modeling performance and domain adaptation, whenever there is a well-formulated test set (and we have well-formulated ways of computing PPLs). We follow previous work on kNN-LM and speculative decoding by employing the same evaluation metrics. For section 6.3, it is an open-ended generation setup, where there is no standard test data, so we mainly rely on human eval for generation quality. We agree that including rigorous human evaluation will enhance the validity of the results, and we are in the process of conducting human evaluation for Sections 6.1 and 6.2.
>
>
> ## Regarding question 2
>
> >**Data Highlighting in Table 1**: There is an issue with the data highlighting in Table 1. The perplexity (ppl) index of KCD-LM in applications in medical and law scenarios is not better than that of the Base model. Does this indicate that using a chunk-based generation method may reduce the quality of generation? Could you please discuss possible reasons for this discrepancy and its implications for the effectiveness of CD-LM in different domains?
>
> Thank you for pointing out the issue with the data highlighting in Table 1. We apologize for the oversight; the shading was missing in the version you received. We have updated the manuscript to include the correct shading and added clarification to prevent any confusion.
>
> To address your concern, we'd like to clarify that our model, **KCD-LM, does outperform the Base LM (137M) in all domains, including the medical and law scenarios**. The confusion may have arisen because the rows for Teacher Model (1.5B) and Base LM fine-tuned were not shaded in the submitted version. These models are included for reference purposes and are not direct competitors to our training-free approach. In the updated version of Table 1, we have shaded these rows to make this distinction clear.
>
>
> ## Regarding question 3
>
> >**Perplexity Calculation**: Regarding the metric PPL: Can you provide the detailed calculation method of PPL? In my understanding, since there is no fixed vocabulary for this kind of method, PPL cannot be calculated.
>
> **In fact this is one of our technical contributions.** For PPL we need to assign probabilities under CD-LM to given sequences. This is non-trivial as you mentioned, since each token in the given sequence could either come from the base LM or from some retrieved chunks, and it becomes a combinatorial problem to enumerate all possible trajectories.
>
> For this exact purpose, we derive an efficient dynamic program (intuition similar to *backward algorithms*) to compute the CD-LM probabilities, described in **Section 5**, with full derivation details and explanations in **Appendix A**,  and a detailed example of perplexity calculation in **Appendix B**.
>
> For a reformulated high-level explanation and a running example, please refer to our response to **Reviewer e9UK (Question 1)**.

---

> ### Comment · Reviewer_78sv · 2024-11-27
>
> Many thanks for the detailed response. I will keep my score unchanged.
>
> A small suggestion: the evaluation metrics adopted in KNN-LM or speculative decoding are far away from satisfactory [1]. So I don't think it is a good choice to follow their evaluation metric. I suggest that the authors should make as much use of human evaluation as possible in future studies, or at least, utilize LLM-based evaluation.
>
> [1] KNN-LM Does Not Improve Open-ended Text Generation

---

> > ### Author Response · Authors · 2024-12-03
> > **Further Evaluations on Text Generation (Part 1)**
> >
> > Thanks for your suggestion! We acknowledge that text generations are hard to evaluate, and we tried our best to incorporate diverse evaluation metrics including LLM-based evaluations to more reliably measure the generation qualities from our approach.
> > **Below we provide a bit more context of our choice of evaluation metrics, and also present new results with LLM-as-a-judge evaluations.**
> >
> > ## 1. Our choice of evaluation metrics
> >
> > - We use PPL as an intrinsic measurement of how well CD-LM, as a proposed new language model with its adjusted distribution with chunk injections, fits the distribution of real in-domain text data in KCD-LM. PPL computation was derived in Section 5, and our focus was to evaluate the distribution shift for domain adaptation.
> >
> > - Nonetheless, for KCD-LM, we acknowledge that relying solely on PPL (as in the original kNN-LM work) does not fully capture the quality of text generation [1]. This is why we **also evaluated the performance of open-text generation with KCD-LM using MAUVE** [2]  evaluation. We chose MAUVE since it also claims, based on empirical comparisons, that it correlates well with human evaluation (Section 4.3 in the MAUVE paper [2]), and **MAUVE was used in [1] too**.
> >
> > - For SCD-LM, our primary focus is on generation efficiency as with self-distillation we do not expect model distribution to improve. We also measured the text generation quality with the **model PPL on sampled generations, similar to [1] (Section 3.3)**.
> >
> > Moreover, During the rebuttal phase, we implemented a verification step for SCD-LM as a straightforward extension to maintain the same generative distribution, ensuring that the generation quality remains unaffected. For detailed results regarding the verification step, please see our [response to Reviewer H44F](https://openreview.net/forum?id=nrvoWOWcyg&noteId=7TKO7K1KWN).
> >
> > In fact, **the evaluation metrics for open-text generation we adopted, such as MAUVE (KCD-LM, Sec 6.1), model PPL on generated texts (SCD-LM, Sec 6.2), and human evaluations (ECD-LM, Sec 6.3), were actually motivated by [1] (Section 3.3)** where they show meaningful assessments to compensate the insufficient measurement purely based on perplexity for kNN-LM.
> >
> >
> > >[1] kNN-LM Does Not Improve Open-ended Text Generation
> >
> > >[2] MAUVE: Measuring the Gap Between Neural Text and Human Text using Divergence Frontiers

---

> > > ### Author Response · Authors · 2024-12-03
> > > **Further Evaluations on Text Generation (Part 2)**
> > >
> > > ## 2. Additional Evaluations using LLM-as-a-judge
> > >
> > > Following your recommendation, we conducted additional evaluations using the **LLM-as-a-judge** approach to provide a more comprehensive assessment of our model's performance in **Sections 6.1 and 6.2**.
> > >
> > > We employed GPT-4o-mini to perform **pairwise comparisons between the outputs of our models and the baselines** (we found GPT-4o-mini shows similar performance with GPT-4o on preliminary evaluations). The evaluation involves presenting a prefix and two continuations generated by different models, and GPT-4o-mini judges which continuation better continues the prefix.
> > >
> > > For **KCD-LM** (Section 6.1), we evaluated 1,000 examples for each domain, following the setup for Table 6.1:
> > >
> > > | **Dataset** | **CD-LM Better** | **Baseline Better** |
> > > |-------------|-------------------|---------------------|
> > > | Code        | 505               | 495                 |
> > > | Medical     | 864               | 136                 |
> > > | Law         | 528               | 472                 |
> > > | Wikitext    | 636               | 364                 |
> > >
> > > For **SCD-LM** (Section 6.2), we evaluated 800 examples from our benchmarking prompts with two base LLMs, using Llama2 and Mistral to generate texts with CD-LM:
> > >
> > > | **Model**                  | **CD-LM Better** | **Baseline Better** |
> > > |----------------------------|------------------|---------------------|
> > > | Llama-2-7b-chat            | 502              | 298                 |
> > > | Mistral-7b-instruct-v0.2   | 456              | 344                 |
> > >
> > > These results indicate that **CD-LM consistently outperforms the baseline models for both KCD-LM and SCD-LM** when judged by GPT-4o-mini.
> > >
> > > Furthermore, inspired by G-Eval [3], we have also conducted a **preliminary fine-grained evaluation**. GPT-4 assessed generated responses based on six aspects: **Relevance**, **Clarity and Organization**, **Accuracy**, **Completeness**, **Language Quality**, and **Creativity and Engagement**. Each aspect was rated on a scale from 1 (very poor) to 5 (excellent), with brief explanations provided. Below are the preliminary results for both Llama-2-7b-chat and Mistral-7b-instruct-v0.2:
> > >
> > > | **Aspect**                | **Llama (Baseline)** | **Llama (SCD-LM)** | **Mistral (Baseline)** | **Mistral (SCD-LM)** |
> > > |---------------------------|----------------------|--------------------|------------------------|----------------------|
> > > | Relevance                 | 4.03                 | 4.06               | 4.31                   | 4.45                 |
> > > | Clarity and Organization  | 4.01                 | 4.15               | 4.36                   | 4.42                 |
> > > | Accuracy                  | 3.33                 | 3.33               | 3.71                   | 3.81                 |
> > > | Completeness              | 3.54                 | 3.85               | 4.04                   | 4.30                 |
> > > | Language Quality          | 4.61                 | 4.66               | 4.79                   | 4.81                 |
> > > | Creativity and Engagement | 3.21                 | 3.21               | 3.48                   | 3.52                 |
> > >
> > > These detailed evaluations show that **CD-LM consistently achieves higher scores in Clarity and Organization, Completeness, and Language Quality for both models, while maintaining similar performance in other aspects**.
> > >
> > > We are currently in the process of completing the fine-grained evaluations with more samples, as well as designing human evaluations similarly. We anticipate to include these LLM-based measurements in the revised manuscript to present a more comprehensive set of evaluations.
> > >
> > > > [3] G-Eval: NLG Evaluation using GPT-4 with Better Human Alignment

---

> > > > ### Author Response · Authors · 2024-12-03
> > > > **Further Evaluations on Text Generation (Part 3)**
> > > >
> > > > We also conducted additional LLM-as-a-judge evaluations to **measure the factuality of ECD-LM's generations (Section 6.3)**, as one goal of ECD-LM is to inject more factual knowledge into its outputs. We find that **for LLaMA-2-7B-Chat and Mistral-7B-Instruct-v0.2, ECD-LM improves factual accuracy**. Please see our [response to reviewer q7qs](https://openreview.net/forum?id=nrvoWOWcyg&noteId=aCOOiDJ8GQ) for the evaluation results.
> > > >
> > > > Thank you again for your constructive feedback. We sincerely hope our explanations and additional results have resolved your questions, but please don’t hesitate to let us know if you have any additional questions or suggestions. We are more than happy to provide further clarifications if needed, as well as incorporating valuable suggestions to further improve our work.

---

### Official Review · Reviewer_q7qS · 2024-11-03

**Soundness:** 3
**Presentation:** 4
**Contribution:** 4
**Rating:** 8
**Confidence:** 4

**Summary:**

This paper proposes a new decoding technique called Chunk-Distilled Language Modeling (CD-LM). The technique could be interpreted as using a novel phrase-level retrieval mechanism to do phrase-level autocompletion, when confident, during decoding. Similar to speculative decoding, this speeds up decoding as a cheaper process has produced the next chunk of tokens, allowing for the base model to skip decoding steps. Unlike speculative decoding this can change the sampling model's output distribution: potentially for the better or to enable novel applications. Similar to retrieval augmentation, this technique improves grounding to an external datastore. Unlike, retrieval augmentation, this does not incur any cost by increasing the context window length.

To do this the authors come up with a novel mechanism to select high confidence / memorized phrases, via the observation that strong LLMs have very high probability for phrases that appear often in their pretraining. We can select these spans to insert in the chunk datastore, which is a collection of Tries. The leaves of the tries contain the chunk's contexts, presumably to update the KV cache with.  Alternatively, the authors also explore manually constructed datastores, that can open up new possibilities for grounding to external sources.

During decoding, the decoded context so far, except for the last token, is encoded into a context vector, to be used to retrieve a relevant trie. The last token in the decoded sequence so far is used as the entry token. The first token of the trie has to match the entry token, to preserve fluency. The similarity of a context vector to a trie's context vector forms the acceptance probability for the decoding algorithm. If accepted, the decoding algorithm fills the sequence with the retrieved chunk, and skipping decoding for that chunk of tokens. The paper also introduces a novel method to compute perplexity given this phrase-level retrieval insertion mechanism.

The authors evaluate this decoding method in three settings: knowledge distillation (a bigger model is used to score and construct the chunk datastore), self distillation (the model itself is used to construct the chunk datastore -- e.g. turning into a sort of cache) , and expert distillation (hand constructed datastores from hand selected datasets, maybe representing new domains or tasks). The authors report improved decoding speeds, while maintaining quality and sometimes improving quality when appropriate, due to the new ability to ground to an external phrase-level datastore.

**Strengths:**

Originality:
- The authors introduce a new decoding paradigm that is quite novel, and has the ability to do fine-grained, phrase-level, grounding to an external source without incurring any extra context length like RAG does, all while potentially saving decoding time by skipping decoding steps. In essence, CD-LM introduces a phrase-level cache with fuzzy matching. This cache essentially gives the model the ability to autocomplete from the cache with some confidence threshold. The flexibility of being able to fill the cache with *anything* is a powerful paradigm, as the authors demonstrate very well in the paper.
- Even if there are short-comings in this paper, I believe this efficient, granular grounding technique will be of value to the community and will inspire future work. In the general LLM case, there are many ways to extend this work in the future to improve it or make it more practical for production. Even as-is, there are specific use cases that would likely benefit from such an approach.

Significance:
- For up to a strong 7B model, the authors demonstrate the ability of the method to save decoding time (10-20%, assuming the cached contexts cover the eval data), while maintaining the overall quality of the model.
- A novel way to ground to external source at the phrase level without training -- especially useful for private information retrieval. On private information retrieval, it outperforms RAG-based few-shot alternative.
- A novel way to distill some performance from a larger model during only decoding, without any online decoding cost. Gets perplexity results similar to the base LM finetuned on the domain, without having to do training.

Quality:
- This is a high quality paper, with many experimental results and detailed descriptions. The results give the reader confidence that within the constraints of their experimental setup, that the method works well. In 6.1, it is demonstrated that a sort of large-to-small model distillation does indeed happen when using a larger model to identify phrases for the cache. In 6.2 it is demonstrated that in self-distillation, the model does maintain a similar output distribution while improving decoding time by 10-20% and reducing forward passes by 20-40%. In 6.3 it is shown that the cache can be manually filled via various methods to inject new information into the model.
- I do not have any issues with the claims made in this paper.

Clarity:
- The writing in this paper is excellent. It is easy to follow and introduces an appropriate amount of notation to make the understanding very clear. I did not find any grammatical or writing errors.
- The paper is well structured to explore various facets of the approach under different use cases.
- Methodology is very well documented, increasing reproducibility.

**Weaknesses:**

1. The limitations of the experiments in this paper should be stated more clearly. Namely:

-  The contexts that are used to construct the chunk datastore / cache aligns well with the evaluation setting in every experiment in the paper. It is not possible to tell what the pitfalls of the retrieval mechanism might be when its pushed to its limit with very large collections of tries. For example in real applications, if CD-LM was used to cache the top 20% of queries with traffic (covering most topics essentially), would there be any quality interference for a tail query? How about as more items are cached and the retrieval space gets denser?
- Regarding this, unlike speculative decoding, CD-LM is essentially building a cache. The decoding time savings then is tied to how well the cache covers real evaluation traffic. In this paper it is always matching and so it represents only good-scenario results for a well built / updated cache.
- The chunk datastores considered in the experiments of this paper are quite small. It's unclear if the speed gains would hold if they were to grow in size, and at what extent the method would break down. Presumably when it stops fitting in host memory.
- There is a hidden cost in having to run LLM scoring to construct the datastore. To get sufficient chunk coverage on real traffic this may be very expensive. However, this mechanism could be used as a straightforward cache of phrases from previous live responses also, avoiding this offline cost.

2. Evaluations in this paper could represent the end model quality more directly. While authors use a variety of metrics that are fine: perplexity, ROUGE, Bleurt, and even human evaluated fluency, the results would be stronger with a more rigorous human evaluation of the overall quality of the text. This could perhaps be represented as a preference based SxS for Section 3.2 experiments.
- Similarly, for "injecting factual knowledge", this grounding capability would much benefit from an evaluation of factuality or response quality itself, rather than just relying on entity distributions. While fluency is important, mis-retrieved entities from context may hurt factuality, but this is not represented in the evaluations.

**Questions:**

Is there any analysis on how much the speed up from skipping tokens during decoding is offset by the trie lookup? How about for collections of tries?

What is the definition of f_theta that is used? How is the pooling from the context into the context vector defined? (Could not find it in the paper.)

If there is no coverage from the chunk datastore, do we expect a speed loss, due to the unnecessary NN search?

---

> ### Author Response · Authors · 2024-11-25
> **response to weakness 1**
>
> Dear reviewer q7qS,
>
> Thank you for your thoughtful review! We really appreciate the amount of time and effort you put into reviewing our paper. Here are our responses:
>
> ## Regarding weakness 1
>
> We agree with your characterization of the limitations and will state them more clearly in the revised version. We greatly appreciate your detailed thoughts and comments, many of which resonate strongly with our intended vision. For instance, your observed that the chunk cache can be built directly "from previous live responses," which precisely reflects the practical application we envision for our approach.
> While we believe that many practical settings are similar to those in which we have tested CD-LM (specifically, where there is a known domain that a limited datastore represents well), the questions of scalability are well worth stating clearly and exploring in future work.

---

> ### Author Response · Authors · 2024-11-25
> **Response to weakness 2 and question 1**
>
> ## Regarding weakness 2 and question 1
>
> Thank you for pointing out these ways of strengthening the results.  We agree and are running these analyses.  We will update the response with the results (assuming they are ready by the end of the discussion period).

---

> ### Author Response · Authors · 2024-11-25
> **Response to question 2**
>
> ## Regarding question 2
>
> >What is the definition of $ f_\theta $ used in the model? How is pooling from the context into the context vector defined? This detail was not found in the paper.
>
> Thank you for bringing this up! It is indeed defined in Section 3.1 in Equation (2) when we define the Transformer forward pass.
>
> In our model, $ f_\theta $ represents the Transformer architecture that processes the input token sequence $ (x_1, x_2, \ldots, x_{n-1}) $. Specifically, $ f_\theta $ maps this sequence to the context vector $ h_n $, which is the last hidden states from the final layer of the Transformer. That is, we do not apply any pooling operations; instead, the context vector $ h_n $ is directly obtained from the last layer's final hidden state. Equation (2) indicates this, as the output $h_n$ from $f_\theta$ is used to compute logits for next token in vocabulary in standard Transformer LM computation.
>
> If this needs more emphasis when we use $f_\theta$ in later sections such as 4.2, we will add a recap to make it explicit and clear.

---

> ### Author Response · Authors · 2024-11-25
> **Response to question 3**
>
> ## Regarding question 3
>
> >If there is no coverage from the chunk datastore, is there an expected speed loss due to unnecessary nearest neighbor (NN) searches?
>
> By structuring our decoding algorithm to **perform LM decoding and NN retrieval in parallel**, we ensure that **unnecessary NN searches do not introduce any speed loss**. Here's how the process works:
>
> 1. Parallel Execution:
>
>    - LM Decoding: At each time step, the LM begins predicting the next token as it normally would.
>
>    - NN Retrieval: Concurrently, we check if a token trie exists for the current token. If it does, we use the hidden state associated with the current token to perform an NN search within that trie.
>
> 2. Outcome Handling:
>
>    - NN Search **Hit**: If the NN search finds a chunk with a similarity exceeding our predefined threshold, we accept this chunk and incorporate it into the output. We then halt the LM decoding for this step, as the chunk provides the necessary continuation.
>
>    - NN Search **Miss**: If the NN search does not find a suitable chunk (i.e., the maximum cosine similarity does not exceed the threshold), we simply proceed with the token predicted by the LM, which has been running in parallel without interruption.
>
> 3. Efficiency Consideration:
>
>    - The NN retrieval process is designed to be faster than or at least as fast as the LM decoding step. This is because the search space within a token trie is smaller than the entire vocabulary over which the LM computes its softmax operation.
>
>    - By running both processes in parallel and only integrating the NN retrieval when it provides a meaningful result, we avoid any additional latency that could be caused by unnecessary NN searches.
>
>
> Implementation-wise,
>
> - We use multithreading to run the LM decoding and the NN retrieval concurrently.
>
> - Upon completion of both processes, we decide whether to use the LM's predicted token or the retrieved chunk based on the NN search outcome.
>
> We will make sure to include this algorithm in our appendix. Thank you for the insightful question!

---

> ### Comment · Reviewer_q7qS · 2024-11-26
>
> Thank you for your clarifications! I remain happy with my rating at this time.

---

> ### Author Response · Authors · 2024-12-03
> **Response to Weakness 2.2**
>
> ## Regarding Weakness 2.2
>
> > For "injecting factual knowledge", this grounding capability would much benefit from an evaluation of factuality or response quality itself, rather than just relying on entity distributions. While fluency is important, mis-retrieved entities from context may hurt factuality, but this is not represented in the evaluations.
>
>
> Thank you for highlighting the importance of evaluating the factuality of the generated text. To address your concerns, we conducted additional experiments to assess the factual accuracy of outputs produced by ECD-LM compared to the baseline models.
>
> To evaluate factual accuracy, we employed GPT-4o to rate each generated text on a scale from 1 to 5, where:
> - 1: Completely inaccurate
> - 2: Mostly inaccurate
> - 3: Partially accurate
> - 4: Mostly accurate
> - 5: Fully accurate
>
> **GPT-4o compared each generation against a verified source document** (the Wikipedia article on Alan Turing), providing both **a score and a detailed rationale**. We generated and evaluated 100 samples for each model in both settings.
>
> The average factual accuracy scores are as follows:
>
> | **Model**                      | **Baseline** | **ECD-LM** |
> |--------------------------------|--------------|------------|
> | LLaMA-2-7B-Chat              | 3.31         | 3.40       |
> | Mistral-7B-Instruct-v0.2     | 3.73         | 3.94       |
>
>
> We find that, **for LLaMA-2-7B-Chat and Mistral-7B-Instruct-v0.2, ECD-LM improves factual accuracy**. The average scores increased from 3.31 to 3.40 and from 3.73 to 3.94, respectively.
>
> These findings suggest that our ECD-LM approach can enhance factual grounding in models. By integrating external chunks, the models produce outputs more aligned with verified information. We are in the process of finalizing the experiments and will include these results and a detailed analysis in the paper.
>
> In the meantime, we also we conducted **LLM-as-a-judge evaluations (including preference based SxS)** in the limited time, and the results showed that KCD-LM (section 6.1) and SCD-LM (section 6.2) consistently outperform the baseline. Please see [our response to reviewer 78sv](https://openreview.net/forum?id=nrvoWOWcyg&noteId=H1avyZntqe) for the evaluation results.

---

### Official Review · Reviewer_H44F · 2024-11-04

**Soundness:** 4
**Presentation:** 4
**Contribution:** 3
**Rating:** 6
**Confidence:** 5

**Summary:**

This paper introduces a chunk-distilled language model (CD-LM), which uses a chunk proposal model (CPM) to retrieve fine-grained chunks and integrate them into the generation process. This approach offers two main advantages: (1) it distills knowledge from a larger model if the CPM datastore is encoded by that larger model, and (2) it saves decoding time by selecting chunks of multiple tokens as completions. The overall decoding algorithm in Section 3.2 is similar to speculative decoding.

The work is technically sound. For example, unlike previous retrieval-augmented generation (RAG) methods that directly search in a vector-based datastore, this model introduces an “entry token” and a trie-tree structure to reduce storage space. Additionally, in Section 5, they use dynamic decoding to marginalize the joint distribution $p(x, z)$, where $x$ represents the sequence and $z$ is the hidden state for accepting or rejecting the retrieved chunks.

**Strengths:**

The proposed CD-LM is well-designed and technically sound. The paper is well-written and easy to follow. The authors also conducted extensive experiments in the appendix, lending additional robustness to their findings.

**Weaknesses:**

While the algorithm in Section 3.2 appears reasonable, it cannot ensure the same properties as speculative decoding, namely that sampling $ x $ from $ p(x) $ is equivalent to sampling from $ q(x) $. In other words, sampling from the chunk proposal model may introduce a distribution shift, potentially reducing performance. The self-distillation experiments deepen this concern: as shown in Figure 6, saving about 20% of mean token time results in a significant performance drop across many LLMs, as measured by BLEURT.

In the knowledge distillation (KD) setting, building the datastore requires two full passes through the corpus, i.e., first with the larger model to select chunks and then with the base model to build the vector datastore. This time cost may be a concern.

**Questions:**

What happens if there are chunks with the same string but different contexts? I assume they would have different hidden representations. Are all of them stored?

---

> ### Author Response · Authors · 2024-11-25
> **Response to weakness 1**
>
> Dear reviewer H44F,
>
> Thank you for your time and effort! We appreciate your positive review and constructive feedback.
>
> ## Regarding weakness 1.1
>
>
> > While the algorithm in Section 3.2 appears reasonable, it cannot ensure the same properties as speculative decoding, namely that sampling $x$ from $p(x)$ is equivalent to sampling from $q(x)$. In other words, sampling from the chunk proposal model may introduce a distribution shift, potentially reducing performance. The self-distillation experiments deepen this concern: as shown in Figure 6, saving about 20\% of mean token time results in a significant performance drop across many LLMs, as measured by BLEURT. The self-distillation experiments deepen this concern: as shown in Figure 6, saving about 20\% of mean token time results in a significant performance drop across many LLMs, as measured by BLEURT
>
> While CD-LM shares similarity with speculative decoding in inference acceleration, **our main focus is not yet another variant of speculative decoding**. Instead, an approach that can both speed up generation and adapt model distributions (to better models or specific domains) is what we wanted. We want to show CD-LM as a flexible framework that can cover various application scenarios. In the case of self-distillation (SCD-LM), we can add the verification step to ensure the distribution is the same to make it exactly another variant of speculative decoding (*but we felt it was a **straightforward extension** and was not the main contribution of the paper*).
>
> We agree that sampling from the chunk proposal model may introduce a distribution shift in self-distillation, especially when the retrieval similarity threshold $\eta$ is set low (less likely to reject a chunk). **To ensure we have the same properties as speculative decoding, we can easily reintroduce the verification step into SCD-LM**. Here are the experimental results for SCD-LM augmented with the verification step under different chunk extraction thresholds $\gamma$:
>
>
> | Model                    | γ     | TTS (%) | FPS (%) |
> |--------------------------|-------|---------|---------|
> | gpt-xl-conversational    | 0.5   | 25.57   | 31.33   |
> | gpt-xl-conversational    | 0.4   | 23.05   | 29.70   |
> | gpt-xl-conversational    | 0.3   | 28.64   | 32.13   |
> | gpt-xl-conversational    | 0.2   | 28.84   | 31.99   |
> | gpt-xl-conversational    | 0.1   | 25.60   | 31.09   |
> | gpt-xl-conversational    | 0.05  | 25.87   | 31.59   |
> | gpt-xl-conversational    | 0.01  | **30.88** | **33.37** |
> | Llama-2-7b-chat          | 0.5   | 3.07    | 10.75   |
> | Llama-2-7b-chat          | 0.4   | 3.60    | 11.24   |
> | Llama-2-7b-chat          | 0.3   | 4.27    | 11.86   |
> | Llama-2-7b-chat          | 0.2   | 4.82    | 12.35   |
> | Llama-2-7b-chat          | 0.1   | 5.68    | 13.09   |
> | Llama-2-7b-chat          | 0.05  | 6.03    | 13.36   |
> | Llama-2-7b-chat          | 0.01  | **6.71** | **13.91** |
> | Mistral-7B-Instruct-v0.2 | 0.5   | -1.93   | 8.47    |
> | Mistral-7B-Instruct-v0.2 | 0.4   | -1.71   | 8.69    |
> | Mistral-7B-Instruct-v0.2 | 0.3   | -1.18   | 9.22    |
> | Mistral-7B-Instruct-v0.2 | 0.2   | -0.68   | 9.69    |
> | Mistral-7B-Instruct-v0.2 | 0.1   | 0.07    | 10.46   |
> | Mistral-7B-Instruct-v0.2 | 0.05  | 0.36    | 10.76   |
> | Mistral-7B-Instruct-v0.2 | 0.01  | **0.97** | **11.28** |
>
>
> Introducing the verification step enables saving 10–-30\% of forward passes while maintaining the LM's distribution. We find that longer chunks in the retrieval datastore (lower $\gamma$) yield better results.
>
>
> However, compared to Table 3 (SCD-LM without verification), there is a reduction in mean token time saved and forward passes saved. We believe this occurs because the original SCD-LM offloads verification to the datastore construction by preselecting confident phrases for caching, allowing direct generation without re-verification. Adding the verification step now introduces additional overhead.
>
> A potential solution that we are currently exploring is a hybrid approach combining verification with direct injection:
>
> - Low Max Cosine Similarity (above retrieval threshold): Apply verification.
>
> - High Max Cosine Similarity: Skip verification, as the chunk likely continues the context effectively.
>
> This mixed strategy aims to optimize performance by reducing unnecessary verification steps when the continuation is already reliable.

---

> ### Author Response · Authors · 2024-11-25
> **Response to weakness 2**
>
> ## Regarding weakness 2
>
> >In the knowledge distillation setting, building the datastore requires two full passes through the corpus, i.e., first with the larger model to select chunks and then with the base model to build the vector datastore. This time cost may be a concern.
>
> We appreciate this concern. On wikitext-103, our method requires:
>
> - Extracting Hidden States: 2 hours on 1 A6000 GPU
>
> - Extracting Probabilities: 5.5 hours on 1 A6000 GPU
>
> - Total GPU Hours: **7.5 hours** (single GPU)
>
> In contrast, fine-tuning GPT-2 Small demands:
>
> - Training Duration: 8 hours on 8 A6000 GPUs
>
> - Hyperparameter Search: ~192 GPU hours
>
> - Total GPU Hours: **~224 hours**
>
> Our method takes less time than a single pass of fine-tuning (while achieving comparable performance), and drastically less time if we take into account the need for hyperparameter search when fine-tuning. Overall, this is a **30-fold reduction** in computational resources for our method over fine-tuning.

---

> ### Author Response · Authors · 2024-11-25
> **Response to question 1**
>
> ## Regarding question 1
>
> >What happens if there are chunks with the same string but different contexts? I assume they would have different hidden representations. Are all of them stored?
>
> Yes, for the same string, all different hidden representations are stored. This is stated in lines 225–226 of our paper: Each node of a trie contains all of the context vectors corresponding to the unique chunk represented by the node.

---

> ### Author Response · Authors · 2024-11-25
> **Thank you!**
>
> Thank you again for your constructive feedback! We will include the experiments regarding SCD-LM with the verification step in our paper if you think it presents a more comprehensive story. Hopefully you would kindly consider further supporting our work to be shared with broader audience. Please let us know if you have any further suggestions; we would be more than happy to make additional improvements. :)

---

> > ### Comment · Reviewer_H44F · 2024-11-27
> > **Response to Authors**
> >
> > Many thanks for the detailed response and the SCD-LM part is interesting. I don't have additional questions.

---

> ### Author Response · Authors · 2024-12-03
> **New Updates**
>
> ## Updates on SCD-LM + Verification Step
>
> This is a follow-up response to address your concern:
>
> > In other words, sampling from the chunk proposal model may introduce a distribution shift, potentially reducing performance. The self-distillation experiments deepen this concern: as shown in Figure 6, saving about 20% of mean token time results in a significant performance drop across many LLMs, as measured by BLEURT.
>
> In a [previous response](https://openreview.net/forum?id=nrvoWOWcyg&noteId=7TKO7K1KWN), we presented some results on incorporating a verification step into SCD-LM. We have conducted additional experiments by setting the chunk extraction threshold $\gamma$ low enough to store longer spans in the datastore. These adjustments have led to improved performance for both LLaMA-2-7b-chat and Mistral-7B-Instruct-v0.2.
>
> | **Model**               | **TTS (%)** | **FPS (%)** |
> |-------------------------|-------------|-------------|
> | LLaMA-2-7b-chat         | 34.40       | 36.95       |
> | Mistral-7B-Instruct-v0.2 | 26.57       | 32.43       |
>
> The updated results **achieve better speedups compared to our initial results reported in the paper and the previous response, without any degradation in quality** (guaranteed by the verification step).
>
> In the meantime, we also conducted **LLM-as-a-judge evaluations** and the results showed that SCD-LM consistently outperform the baseline. Please see [our response to reviewer 78sv](https://openreview.net/forum?id=nrvoWOWcyg&noteId=H1avyZntqe) for the evaluation results.
>
> We hope that these additional results further address your concerns on SCD-LM.
>
> Thank you once again for your valuable feedback!

---

### Author Response · Authors · 2024-12-04
**Summary of Rebuttal**

We thank all the reviewers for their thoughtful feedback and positive evaluations of our paper!

During the rebuttal phase, we addressed several misunderstandings, clarified key differences from prior works, and incorporated additional experiments as requested.

## 1. Clarifications and Addressing Reviewer Concerns:

- ***Distribution Shift and Verification Step (Reviewer H44F)***: We acknowledged **concerns about potential distribution shift in the self-distillation setting**. To address this, we **introduced a verification step** to ensure that our method maintains the same distribution as the base model, similar to speculative decoding. We provided [new experimental results](https://openreview.net/forum?id=nrvoWOWcyg&noteId=f8YpnTU4jI) demonstrating that **this adjustment preserves performance while achieving better speedups**.

- ***Scalability and Efficiency (Reviewers e9UK)***: We clarified how **CD-LM mitigates storage and computational overhead compared to kNN-LM** by selectively storing high-probability chunks, resulting in a significantly smaller datastore. We explained that our retrieval process is more efficient due to constraints like the entry token, making CD-LM more scalable and practical.

- ***Perplexity Calculation and Technical Contributions (Reviewer e9UK)***: We clarified that our paper includes detailed and intuitive explanations of our perplexity calculation in Appendices A and B. To further address the reviewer’s feedback, we have now incorporated **a new reformulation** of the perplexity calculation in the main paper, providing **a more thorough breakdown of the example from Appendix B**.

## 2. Differentiation from Prior Works:
- ***Copy Is All You Need (Copy Generator) (Reviewer 78sv)***: We clarified that CD-LM differs from the Copy Generator by using a much smaller datastore, achieving faster inference, and requiring no additional training. CD-LM focuses on both speeding up generation and adapting the model's distribution, which is not addressed by the Copy Generator.
- ***Nearest Neighbor Speculative Decoding (NEST) (Reviewers 78sv)***: We explained that while NEST modifies the token-level distribution through interpolation and uses speculative decoding, CD-LM introduces a formal probabilistic framework integrating variable-length chunks, altering the language model's distribution in a principled way. This ensures perplexity fully accounts for chunk-level modifications, which is not the case with NEST.

## 3. Additional LLM Evaluation Results:
- ***LLM-Based Evaluations of Generations in Sections 6.1 and 6.2 (Reviewers q7qS and 78sv)***: In response to requests for more rigorous evaluations, we conducted **LLM-as-a-judge pairwise assessments** (using GPT-4o-mini) and **fine-grained evaluations** (using GPT-4o). [The results](https://openreview.net/forum?id=nrvoWOWcyg&noteId=H1avyZntqe) showed that CD-LM consistently outperforms baseline models. These evaluations strengthen our claims about the quality improvements achieved by CD-LM.
- ***Factual Accuracy Improvement (Reviewer q7qS)***: Using GPT-4o as a fact-checker, [results](https://openreview.net/forum?id=nrvoWOWcyg&noteId=aCOOiDJ8GQ) show that **CD-LM enhances the factual grounding** of generated outputs.

We believe these additions and clarifications further improve the quality and impact of our paper. Once again, we greatly appreciate the reviewers' insights, which helped us improve our work!

---

### Meta-Review · Area_Chair_LJeh · 2024-12-20

**Metareview:**

The paper presents a chunk-distilled LM, where the authors incorporate retrieved chunks in the generation phrase. The chunk knowledge can be distilled by self-distillation, from larger models, or from experts.

In general, the paper does not present much novelty, as it follows previous copy/retrieval-based LMs and the main difference is using chunks as the granularity. That being said, reviewers generally believe this is a solid paper with comprehensive experiments.

**Additional Comments On Reviewer Discussion:**

Reviewers are generally on the positive sides, although most are borderline positive.

---

### Decision · Program_Chairs · 2025-01-22

Accept (Poster)